# Nonvesicular lipid transfer drives myelin growth in the central nervous system

Jianping Wu ®[1,2,3], Georg Kislinger ®[1,2], Jerome Duschek ®[4,5], Ayşe Damla Durmaz[1,2], Benedikt Wefers ®[2], Ruoqing Feng[1,2,3], Karsten Nalbach ®[4,5], Wolfgang Wurst ®[2,5,6,7], Christian Behrends ®[4,5], Martina Schifferer ®[2,5] & Mikael Simons ®[1,2,5,8] ✉

Oligodendrocytes extend numerous cellular processes that wrap multiple times around axons to generate lipid-rich myelin sheaths. Myelin biogenesis requires an enormously productive biosynthetic machinery for generating and delivering these large amounts of newly synthesized lipids. Yet, a complete understanding of this process remains elusive. Utilizing volume electron microscopy, we demonstrate that the oligodendroglial endoplasmic reticulum (ER) is enriched in developing myelin, extending into and making contact with the innermost myelin layer where growth occurs. We explore the possibility of transfer of lipids from the ER to myelin, and find that the glycolipid transfer protein (GLTP), implicated in nonvesicular lipid transport, is highly enriched in the growing myelin sheath. Mice with a specific knockout of *Gltp* in oligodendrocytes exhibit ER pathology, hypomyelination and a decrease in myelin glycolipid content. In summary, our results demonstrate a role for nonvesicular lipid transport in CNS myelin growth, revealing a cellular pathway in developmental myelination.

Information processing in complex organisms relies on rapid nerve conduction, facilitated by myelin covering axons[1]. Myelination takes place not only during development but also in adulthood, where it is associated with adaptive responses within neuronal circuits like those underlying learning and memory[2–6]. Moreover, deficiencies in remyelination, as seen in diseases such as multiple sclerosis, can result in profound neurological deficits[7]. Hence, achieving a thorough understanding of the cell biology underlying myelination is an essential objective. Oligodendrocytes synthesize myelin in the CNS, exhibiting substantial biosynthetic capacity to generate up to 50 myelin sheaths with multiple layers of membrane[4,8,9]. This process places a significant demand on the machinery responsible for protein and lipid synthesis. Myelin has a unique composition, with 70–80% being lipids, including

cholesterol, plasmalogens and galactosylceramide (GalCer), and a small set of proteins such as myelin basic protein (MBP) and proteolipid proteins (PLP)[1]. The intricate process of synthesizing and transporting large amounts of specialized membrane to myelin remains an unanswered question.

The current model of myelin biogenesis implies that myelin outgrowth occurs by the wrapping of the leading edge, the inner tongue of myelin, around the axon beneath the previously deposited membrane[10–13]. There is an elaborated network of cytoplasmic channels within compacted myelin that serve as potential routes for vesicular transport of membrane to the growth zone at the inner tongue in the developing myelin sheath[12,14–16]. By using vesicular stomatitis virus glycoprotein G as a reporter, we have previously

[1]Institute of Neuronal Cell Biology, Technical University of Munich, Munich, Germany. [2]German Center for Neurodegenerative Diseases, Munich, Germany. [3]Graduate School of Systemic Neurosciences, LMU Munich, Munich, Germany. [4]Medical Faculty, Ludwig-Maximilians-University München, Munich, Germany. [5]Munich Cluster for Systems Neurology (SyNergy), Munich, Germany. [6]Institute of Developmental Genetics, Helmholtz Center Munich, Neuherberg, Germany. [7]Chair of Developmental Genetics, Munich School of Life Sciences Weihenstephan, Technical University of Munich, Freising, Germany. [8]Institute for Stroke and Dementia Research, University Hospital of Munich, LMU Munich, Munich, Germany. ✉e-mail: mikael.simons@dzne.de

been able to follow membrane protein trafficking within the forming myelin sheath and defined the inner tongue as the growth zone[12]. Moreover, recent work demonstrate that VAMP2/3-mediated vesicular transport in oligodendrocytes is indispensable for oligodendrocyte development and myelin formation[17–20]. This process would require transporting molecules from the endoplasmic reticulum (ER) to the Golgi apparatus, located in the cell body and proximal processes[21], followed by vesicle budding from the Golgi and transporting these vesicles through the cytoplasmic channels into myelin, and the subsequent fusion of these vesicles with target membranes[17,22–25]. Despite evidence supporting vesicular membrane trafficking, the slow process prompts the hypothesis that nonvesicular lipid transfer from the ER may contribute. Nonvesicular lipid transport plays a crucial part in intracellular lipid trafficking, and when mediated by lipid-transfer proteins, it is a rapid process for transport of different lipid species between membranes[26–29]. Because the most abundant lipids found in myelin, including its glycolipid, galactosylceramide, are synthesized in the ER without undergoing modification in the Golgi apparatus[30], we tested whether nonvesicular lipid transport plays a role in lipid transfer from the ER to myelin, a specialized domain of oligodendroglial plasma membrane. Here, we show that the tubular ER and the nonvesicular lipid transfer protein, GLTP, are highly enriched in the growing myelin sheath. Targeted knockout of *Gltp* in oligodendrocytes display ER abnormalities, hypomyelination, and reduced myelin glycolipid content. In conclusion, our findings highlight the role of nonvesicular lipid transport in CNS myelin development, uncovering a cellular mechanism crucial for developmental myelination.

## Results

### Tubular ER is enriched in developing myelin and contacts the myelin membrane

We initiated our investigation by characterizing the presence of ER in developing myelin, focusing on mice at postnatal day 14 (P14), a stage at which active myelination occurs. We acquired a volume electron microscopy dataset from P14 mouse optic nerve using ATUM-SEM (Automated Tape-collecting Ultramicrotome-Scanning Electron Microscopy)[31]. Based on the continuity, shape, lumen brightness and membrane appearance[32,33], we identified the ER within the inner tongue, the growing front of myelin (Fig. 1a, Supplementary Movie 1, Supplementary Fig. 1a). Subsequent 3D reconstruction unveiled an ER network comprising mainly tubular ER within the inner tongue. Sometimes the ER appears "lumenless" (Supplementary Fig. 1a), reminiscent of the thin ER observed in neuron[34,35]. We quantified the presence of membrane-bound organelles in developing myelin of three mice at P14 and found that the ER was the most abundant organelle. 82% of P14 myelin have at least one ER, which far exceeded the total of other membrane-bound organelles (20%) (Fig. 1b, c, d, Supplementary Fig. 1b). To confirm this finding, we selected tubular ER markers based on published RNA sequencing data (Supplementary Fig 1c) and conducted immunohistochemistry on spinal cord cross-sections from P14 mice. The myelin marker MBP manifested as distinctive rings, characteristic of myelin structures. Tubular ER markers, receptor expression-enhancing protein 5 (REEP5), reticulon 4 (RTN4) and reticulon 1 (RTN1)[36,37], appear as puncta on MBP⁺ rings (Fig. 1e, Supplementary Fig 1d). Quantification from three mice revealed that 82% of MBP⁺ rings have overlapping REEP5 puncta, 73% have RTN4 puncta, 81% have RTN1 puncta (Fig. 1f, Supplementary Fig 1e). Despite the inner tongue being a compact compartment, oligodendrocytes invest a substantial amount of space for the ER, as inner tongue ER occupied significantly more space than axonal ER (Supplementary Fig. 1f). Additionally, the oligodendrocyte ER was found in close proximity to the plasma membrane of developing myelin (Fig. 1g, h), with 83% of ER-plasma membrane distances being less than 20 nm – a distance feasible for nonvesicular lipid transport[38,39]. In contrast, only

16% of axonal ER is less than 20 nm from axonal plasma membrane (Fig. 1h).

### Tubular ER is associated with active myelination

To determine whether the expression dynamics and subcellular distribution of tubular ER markers correlate with active myelination, we performed immunohistochemistry with antibodies against BCAS1 and MAG, a marker of pre- and actively myelinating oligodendrocytes, and focused on P14 mouse cortex, where these cells are easier to visualize. In the mouse cortex (Fig. 2a, b), myelinating oligodendrocytes (indicated by arrows, BCAS1⁺MAG⁺ cells) showed enhanced expression of tubular ER marker REEP5 and RTN4, compared to premyelinating oligodendrocytes (indicated by arrowheads, BCAS1⁺MAG⁻ cells). Quantification from three P14 animals reveals that 93% of myelinating oligodendrocytes have abundant REEP5, and 82% have abundant RTN4 (Fig. 2c). In contrast, low expression of these tubular ER proteins is observed in premyelinating oligodendrocytes, suggesting that oligodendrocytes increase the expression of tubular ER proteins when myelination starts. Similar observations are made in the cerebellum (Supplementary Fig. 2a-c). Furthermore, in 6-month-old mice with mature myelin, a decrease in tubular ER in the white matter of the spinal cord is noted compared to P14 (Fig. 2d, f). Quantification from three animals at each time point indicates a significant reduction in tubular ER fluorescence signal normalized by myelin signal (Fig. 2e, g). This was consistent with EM analysis, which showed an almost complete absence of the inner tongue with associated ER in adult myelin (Supplementary Fig. 2d, e).

The use of mouse oligodendrocyte culture allows for the examination of the subcellular distribution of various ER subdomains. SEC61b and KDEL, which indicate rough ER and ER sheet, respetively[40], predominantly localize to the cell body. In contrast, tubular ER markers RTN4 and RTN1[37] exhibits a more widespread distribution. Calculating the ratio of peripheral signal to cell body signal reveals that tubular ER surpasses other ER types, indicating that oligodendrocytes extend their ER as tubules toward the growing front (Fig. 2h). Knocking down *Rtn4* using RNAi in primary oligodendrocytes results in a significantly reduced MBP⁺ sheet size (Supplementary Fig. 2f, g). This indicates that tubular ER has a role in the process of membrane extension, which is consistent with a delay in myelination in mice deficient in RTN4[41].

### Mice deficient in glycolipid transfer protein have ER pathology and hypomyelination

We then explored the potential for nonvesicular lipid transport from the ER to the myelin membrane during myelin development. We analysed transcriptome and proteome databases to identify lipid transfer proteins enriched in oligodendrocytes, leading us to recognize glycolipid transfer protein (GLTP) as a promising candidate[42,43]. GLTP stands out as a highly abundant and specific transcript and lipid transfer protein in oligodendrocytes (Supplementary Fig. 3a, b). Through immunohistochemistry, we corroborated GLTP as a protein that distinctly localizes to oligodendrocytes. Co-staining with BCAS1 and MAG revealed higher abundance in myelinating oligodendrocytes compared to the pre-myelinating stages (Fig. 3a, b), with its expression diminishing as myelin matures (Fig. 3c, d). A closer look at P14 myelin displayed focal staining of GLTP within myelin (Fig. 3e, f). We examined the subcellular localization of GLTP using primary oligodendrocyte culture, and observed that GLTP was mainly concentrated in the cell body but also appeared in the cell processes and at the border of MBP⁺ sheet, in contrast to the ER sheet marker, KDEL, which localized primarily in the cell body (Fig. 3g, h).

Myelin contains a large amount of glycolipids, primarily galactosylceramide (GalCer), which is synthesized in the ER, and a small quantity of its derivative, sulfatide, that are synthesized in the Golgi

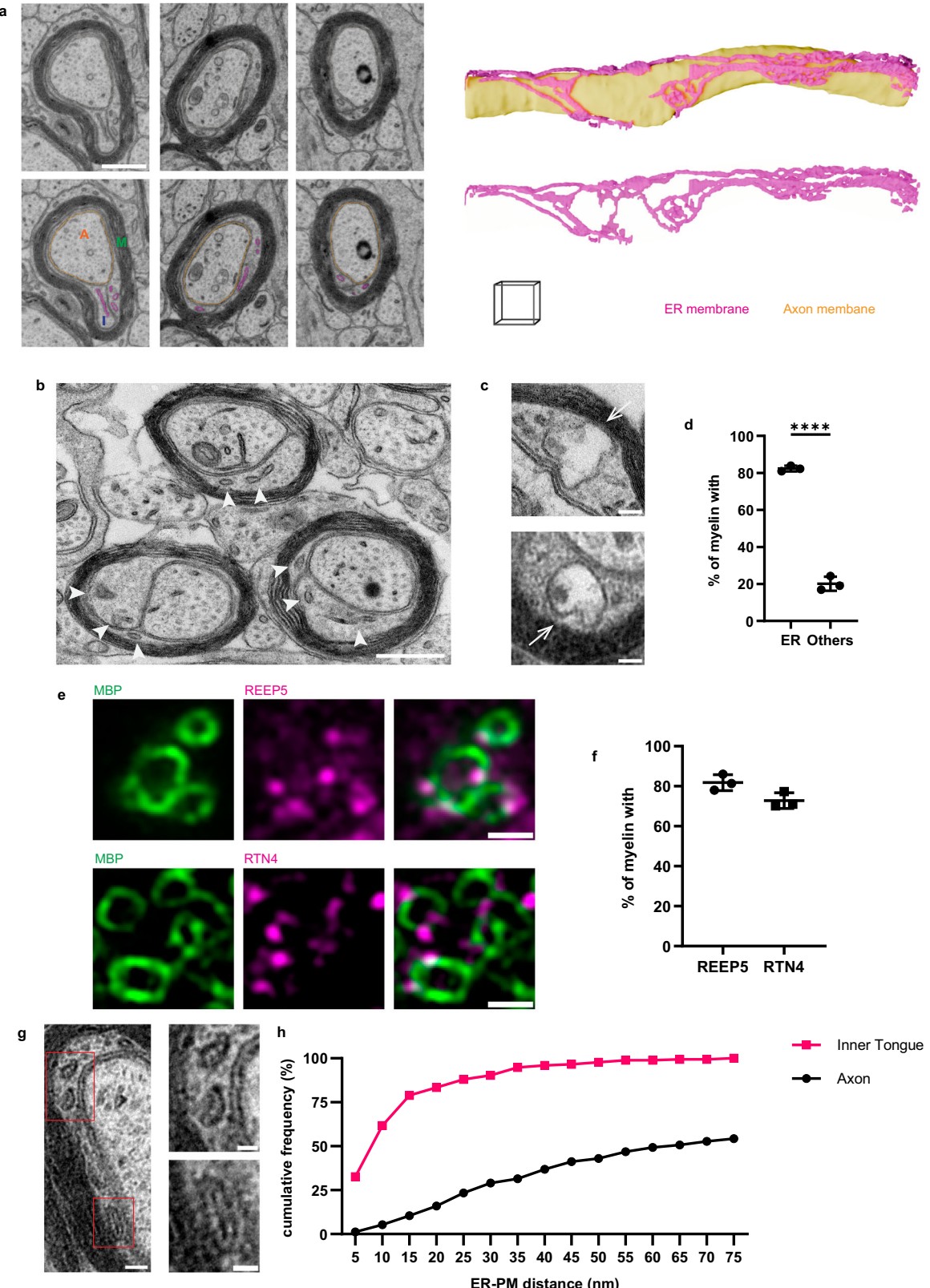

apparatus[44]. We hypothesized that GLTP transfers GalCer from the ER to myelin. To test this hypothesis, we created a whole-body *Gltp* knockout and a line with floxed alleles for conditional deletion of *Gltp* in oligodendrocytes (Supplementary Fig. 4a). Because the constitutive knockout of *Gltp* proved to be embryonically lethal, we continued our analysis with the conditional knockout, which we generated by crossing the *Gltp* floxed mice with *Cnp-Cre* to target cells of the

oligodendrocyte lineage[45]. *Gltp* cKO (*Cnp-Cre, Gltp*^fl/fl^) resulted in reduced GLTP levels in various brain regions, as shown by Western blot and immunohistochemistry analyses (Supplementary Fig. 4b, c). Co-staining with markers for oligodendrocytes confirmed the effective deletion of *Gltp* from these cells.

We started by investigating the impact of Gltp deletion on myelin ultrastructure. *Gltp* cKO mice were capable of forming compact

**Fig. 1 | Tubular ER is enriched in developing myelin and contacts the myelin membrane. a** Volume EM (ATUM-SEM) analysis of developing myelin from postnatal day 14 (P14) mouse optic nerve. Different compartments are marked on the bottom left image: axon (A), myelin (M) and inner tongue (I). ER membrane and axon membrane are segmented, and labeled in magenta and yellow, respectively. 3D reconstruction of ER network is shown together with (upper) or without (bottom) axon membrane, from an 11.25 μm-thick stack (226 slices with 50 nm interval, scale cube 1x1x1 μm). **b** Representative image of the TEM dataset for quantification of membrane-bound organelles in the inner tongue. Arrow heads: ER. **c** Examples of other membrane-bound organelles (arrows). **d** Ratio of inner tongues containing specified organelles, showing mean ± SD, ER: 82.36 ± 1.51, others: 20.13 ± 3.80 ($n$ = 3 wildtype P14 mice, two-tailed unpaired t-test, $t$ = 26.37, df = 4, ****$p$ < 0.0001).

**e** Immunohistochemistry of developing myelin from P14 mouse spinal cord shows tubular ER markers (REEP5 and RTN4) appear as puncta and overlap with MBP marked myelin. **f** Ratio of myelin (MBP⁺ ring) overlapping with specified tubular ER marker. Quantification of (**e**) showing mean ± SD, REEP5: 81.80 ± 3.97, RTN4: 72.79 ± 3.91 ($n$ = 3 wild-type P14 mice). **g** An example with zoom-in views showing short distance between the ER and myelin membrane. **h** Distribution of inner tongue and axonal ER's distance from the plasma membrane. For inner tongue ER, 62% ≤ 10 nm, 83% ≤ 20 nm, 90% ≤ 30 nm ($n$ = 175 ER from three wild-type P14 mice). For axonal ER, 5% ≤ 10 nm, 16% ≤ 20 nm, 29% ≤ 30 nm ($n$ = 337 ER from three wild-type P14 mice). Scale bars: 0.5 μm (**a**), 0.5 μm (**b**), 0.1 μm (**c**), 1 μm (**e**), 100 nm (g left), 50 nm (g zoom-in). Source data are provided as a Source Data file.

myelin; however, we observed intriguing membrane rings within the inner tongue of P14 spinal cord (Fig. 4a) and optic nerve myelin (Supplementary Fig. 5a). Such membrane rings were never observed in *Cnp-Cre* littermates (quantification based on more than 1500 myelin cross-sections for each condition) (Fig. 4b). The thickness of the rings varied, with some showing fewer wraps (indicated by an arrow), others displaying more wraps (indicated by an arrowhead), and some accompanied by a comet tail-like structure (indicated by a star) (Fig. 4a zoom-in). These ring structures were also present in the outer tongue, the cellular process of oligodendrocytes (double arrowhead, Fig. 4a zoom-in), the cell body cytoplasm (Supplementary Fig. 5b) and the nuclear envelope (Supplementary Fig. 5c). Volume electron microscopy (ATUM-SEM) revealed that these rings were long tubes, traceable for up to 11 μm (Supplementary Movie 2). 3D reconstruction illustrated their origin as ER, transitioning into a "lumenless" ER, and eventually rolling up to form a tube (Fig. 4c, Supplementary Movie 3, 4). Moreover, myelin with rings had reduced or no ER. In instances where there were rings in the inner tongue, there was a lower chance of observing the ER, while inner tongues without the rings showed a normal likelihood of observing the ER, similar to the wild type (Fig. 4d). These rings are big and often fill the entire inner tongues, making them close to the myelin (Supplementary Fig. 5d).

How does the *Gltp* knockout and the presence of membrane rings affect myelination? We further examined P28, a time point when the majority of myelination has terminated. *Gltp* cKO exhibited hypomyelination, evident from a decreased g-ratio (inner myelin diameter to outer myelin diameter ratio) (Fig. 5a, b, c). Additionally, myelin inner tongues were swollen in the *Gltp* cKO (Fig. 5d), possibly due to pathology caused by membrane rings. Dysmyelination features like myelin whorls, but not outfoldings, were also increased in the Gltp cKO (Fig. 5e, f, Supplementary Fig. 6a). However, the total number of axons and the percentage of myelinated axons was not significantly lower in the Gltp cKO, suggesting that the major impact of the knockout is on the myelination process (Supplementary Fig. 6b, Fig. 5g). The knockout continues to affect myelin in adulthood, as *Gltp* cKO mice show significantly increased myelin alterations compared to wild-type mice at 3 months old (Supplementary Fig. 6c, d). Together, these observations show that *Gltp* deletion in oligodendrocytes leads to aberrant ER structure and hypomyelination.

### Delivery of glycolipid to myelin is impaired in *Gltp* mutants

To test if GLTP is required for glycolipid transport during myelination, we purified myelin from *Gltp* cKO and *Cre* control mice using the standard two-round sucrose density gradient centrifugation protocol at early (P14) and late (P28) developmental stages of myelination[46]. Subsequently, we conducted a shotgun lipidomics analysis, where GalCer and glucosylceramide were detected together as monohexosylceramide (HexCer) (Supplementary Dataset 1). Principal Component Analysis (PCA) revealed clear separation among the four conditions (Fig. 6f). The volcano plot resulting from multiple t-tests of *Gltp* cKO to Cre control is presented, with the top 10 decreased and increased lipid species listed (Fig. 6a, b, c). HexCer constitute the

majority of the decreased lipids. The decrease is particularly pronounced in HexCer species with normal fatty acid (NFA), whereas those with 2-hydroxy fatty acid (HFA) show no significant changes in *Gltp* cKO (Fig. 6d). This might be explained by the observed higher levels of HFA-glucosylceramide in GalCer-deficient mutants[47,48]. When species were grouped into classes for two-way ANOVA analysis followed by Tukey's post-hoc test, the results showed that *Gltp* cKO altered myelin lipid composition by decreasing HexCer and ether-linked phosphatidylethanolamine (PE O-) while increasing cholesterol, and exhibiting up-and-down changes of phosphatidylserine (PS) (Fig. 6e). Further lipidomics report can be found in Supplementary Fig. 7a-p. To complement the lipidomics findings, we assessed the delivery of GalCer using anti-GalCer antibodies in 2D oligodendrocyte culture capable of forming MBP⁺ sheets akin to myelin[30]. Extracellular and intracellular GalCer were visualized by immunocytochemistry before and after Triton X-100 permeabilization (Fig. 6g). The extracellular to intracellular GalCer ratio was lower in oligodendrocytes derived from the *Gltp* cKO mice, indicating a deficit in GalCer delivery (Fig. 6g, h). Furthermore, the use of *Gltp* siRNA as a complementary approach for gene disruption confirmed the deficiency in GalCer delivery (Fig. 6g, i). The specificity of the anti-GalCer antibody, knockout, and knockdown efficiency were validated (Supplementary Fig. 8a-c). Moreover, GLTP cKO oligodendrocyte cultures have fewer mature oligodendrocytes compared to wild-type cultures (Supplementary Fig. 8d-e) GLTP might have functions beyond non-vesicular lipid transport in myelin[49]. We conducted mass spectrometry-based proteomics on myelin, validating the sample quality through principal component analysis, cellular component GO term enrichment analysis, and GLTP reduction in cKO (Supplementary Fig. 9a-c, Supplementary Dataset 2, 3). The majority of the myelin proteins, including transmembrane proteins transported via the ER-Golgi vesicular pathway, remained unchanged (Supplementary Fig. 9c-e). Enrichment analysis[50] revealed upregulated pathways related to neurodegeneration and lipid droplet organization in GLTP cKO (Supplementary Fig. 9f). Together, our lipidomics and cell biological transport assay provide evidence for the role of GLTP in transferring GalCer to developing myelin.

## Discussion
Myelination involves wrapping the leading edge around the axon, beneath the previously deposited membrane, together with the lateral extension of the different myelin membrane layers towards the nodal regions[10–13]. This model implies that newly synthesized membrane material must traverse the developing myelin sheath to reach the innermost layer adjacent to the axon, through a system of cytoplasmic channels that create a helical path for transporting vesicles to the growth zone at the leading edge. We now demonstrate the existence of an elaborate network of tubular ER that extends into the cytoplasmic channels, reaching all the way to the inner tongue. The ER is ideally positioned to drive myelin biogenesis given its crucial role in synthesizing the bulk of cellular lipids. Signals originating both externally and internally may guide ER-localized enzymes to furnish myelin with membrane lipids.

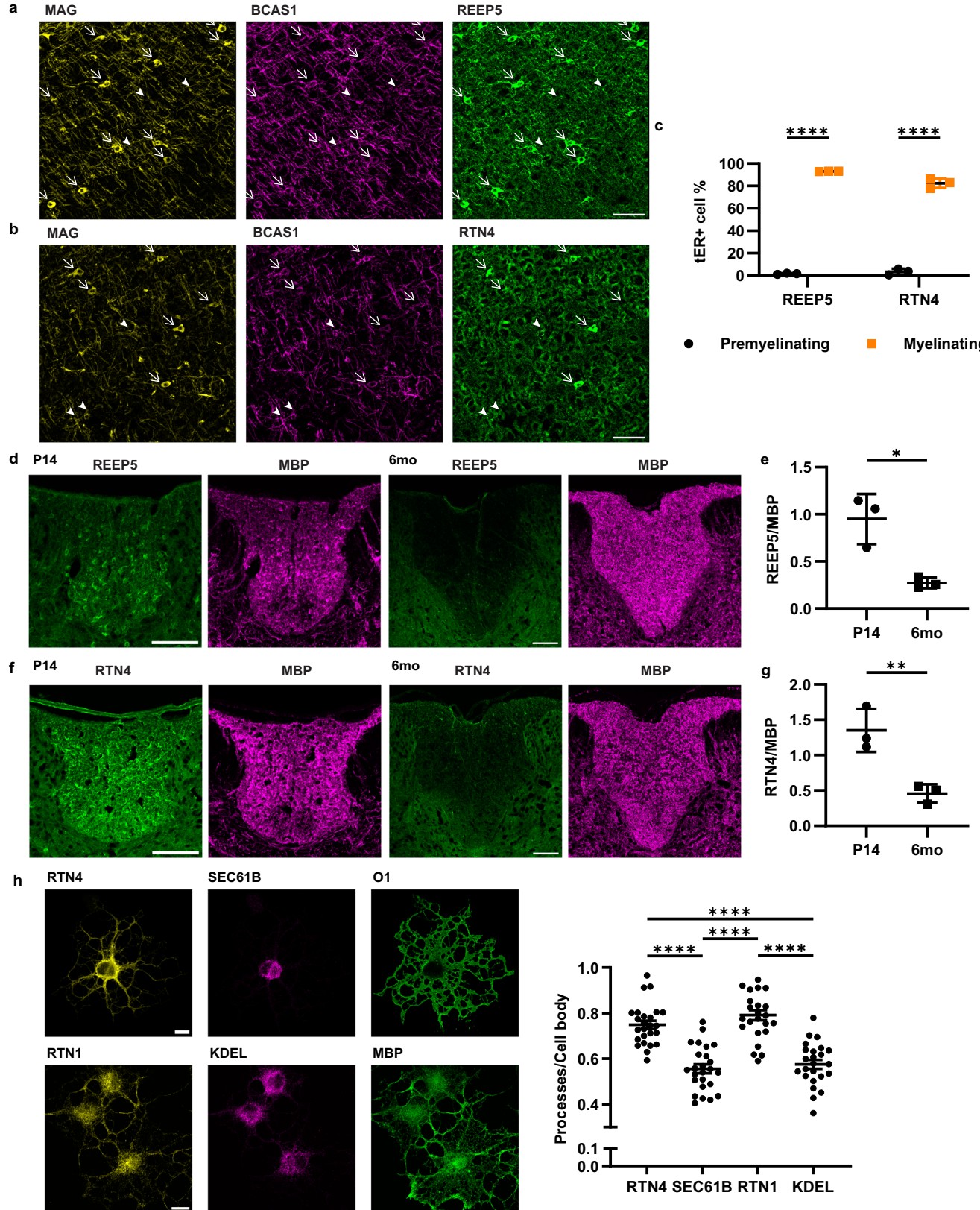

GalCer is a major myelin lipid that is synthesized in the ER[30]. It has been shown that lipid transfer protein GLTP transfers glycolipids, including GalCer between liposomes[51]. In cell lines, GalCer transport persists in the absence of vesicular trafficking and *Gltp* overexpression stimulates GalCer transport[52]. The expression of *Gltp* correlates with myelination[53,54], but whether *Gltp* is required for myelination and the underlying mechanism remain unknown. In this study, we observed a significant enrichment of GLTP in developing myelin. Mutant mice, lacking GLTP in oligodendrocytes, demonstrated a reduction in myelin glycolipid content, and exhibited ER pathology in myelin, hypomyelination and dysmyelination. Our findings suggest that nonvesicular lipid transfer from the ER

**Fig. 2 | Tubular ER is associated with active myelination. a, b** Immunohistochemistry of P14 mouse cortex. Arrow heads: pre-myelinating oligodendrocytes (MAG·BCAS1⁺); arrows: myelinating oligodendrocytes (MAG⁺BCAS1⁺).
**c** Quantification of (**a**) and (**b**) showing mean ± SD. 1.65 ± 0.55% or 3.54 ± 2.60% premyelinating cells are REEP5⁺ or RTN4⁺, 93.04 ± 0.25% or 82.34 ± 4.16% myelinating cells are REEP5⁺ or RTN4⁺ ($n = 3$ wild-type P14 mice, two-way ANOVA followed by Sidak's multiple comparison test, ****$P < 0.0001$). **d, f** Immunohistochemistry of P14 and six-month-old (6mo) adult mouse thoracic spinal cord, focusing on dorsal white matter **e, g** quantification of tubular ER (REEP5 or RTN4) fluorescence signal normalized by myelin (MBP) signal, in the dorsal white matter of thoracic spinal cord, showing mean ± SD, (**e**) P14: 0.95 ± 0.27, 6-month-old: 0.27 ± 0.06 ($n = 3$ wildtype mice for each time point, two-tailed unpaired t-test,

t = 4.302, df=4, *$p = 0.0126$) (**g**) P14: 1.35 ± 0.30, 6-month-old: 0.46 ± 0.13 ($n = 3$ wildtype mice for each time point, two-tailed unpaired t-test, $t = 4.677$, df=4, **$p = 0.0095$) **h** Left: Immunocytochemistry of primary oligodendrocyte culture for tubular ER markers (RTN4 and RTN1), rough ER marker SEC61B, ER sheet marker KDEL[40] and oligodendrocyte marker O1 and MBP for outlining the cells. Right: The ratio of fluorescence signals from different ER subtypes at the cellular processes to that at the cell body, showing mean ± SD, KDEL: 0.58 ± 0.10, RTN1: 0.79 ± 0.11, SEC61B: 0.56 ± 0.10, RTN4: 0.75 ± 0.09 ($n = 24$ cells for KDEL/RTN1, 25 cells for SEC61B/RTN4, one-way ANOVA, F (3, 94) = 36.09, Tukey's post-hoc test: **** $p < 0.0001$) Scale bars: 50 μm (**a, b**), 100 μm (**d, f**), 10 μm (**h**). Source data are provided as a Source Data file.

contributes to myelin biogenesis (Fig. 7a), and in conjunction with vesicular exocytosis, it propels membrane expansion, driving the elongation of the myelin sheath.

One crucial question revolves around explaining the GLTP cKO phenotype. In contrast to mouse models with a disrupted gene encoding UDP-galactose:ceramide galactosyltransferase (CGT), the enzyme critical for GalCer biosynthesis, these models exhibit a loss-of-function phenotype that differs from the observations in this study. CGT-deficient animals generally develop myelin that is slightly thinner but largely normal, with the primary pathology evident in the paranodal loops of the myelin[47,48]. On the contrary, in *Gltp* cKO, we propose a gain-of-toxic function phenotype, possibly triggered by the accumulation of GalCer in the ER. The headgroups of GalCer have been shown to interact with each other via carbohydrate-carbohydrate interactions, potentially involving hydrogen bonds and hydrophobic interactions, leading to the adherence of GalCer bilayers in vitro[55,56]. As a result, GalCer in the lumenal leaflet aids in zipping up the ER, possibly resulting in the observed 'lumenless' phenotype in the mutant. Additionally, due to the accumulation of GalCer at the cytoplasmic leaflet, the ER adopts a rolled-up configuration. The ER has a tendency to roll up when its membrane can interact, as observed, for instance, with non-monomeric GFP on the ER[57] (Fig. 7b). Our work also provides insights into the question of why nature chose GalCer for myelin. It has been postulated that glycolipids are chosen for myelin because of their adhesive property, but the majority of them are synthesized in the Golgi apparatus[58]. GalCer is an exception; it is produced in the ER at the growing front and can be transferred directly to the myelin membrane, making it an ideal glycolipid for myelin.

The molecular details regarding the operation of the nonvesicular machinery are yet to be determined. Nonvesicular lipid transport typically takes place at membrane contact sites, where lipid transfer proteins solubilize lipids, shielding them from the aqueous phase as they move between membranes. GLTP is a cytosolic protein known to interact with vesicle-associated membrane proteins (VAP-A)[59], a protein implicated in the formation of various ER-organelle contact sites[28]. Nevertheless, non-canonical functions of GLTP cannot be excluded[49]. GLTP can function at a VAP-A mediated ER-myelin membrane contact site if one VAP-A subunit[60] binds to a myelin membrane protein while the other binds to GLTP. However, it is also plausible that myelin represents a unique scenario where nonvesicular lipid transport takes place without the need for a tether. The inner tongue, being a small compartment, ensures consistent proximity between the ER and the myelin membrane. Close examination along the inner tongue reveals a continuous association between the ER and the myelin membrane. This suggests that myelin might resemble a situation akin to a test tube with a high concentration of liposomes, where lipid transfer can occur without a tethering mechanism.

An additional unresolved question relates to the transbilayer movement of GalCer across the myelin membrane. GalCer is initially formed in the luminal leaflet but gains rapid access to the cytoplasmic leaflet[61–63]. GLTP facilitates the transfer of GalCer from the ER to the

myelin membrane, where an as-yet-unidentified machinery, potentially involving ATP-binding cassette (ABC) transporters[63], transports GalCer from the intracellular leaflet to the extracellular leaflet. In this scenario, GalCer concentration in the intracellular leaflet of the myelin membrane remains low, creating a gradient and determining the transfer direction as ER-to-myelin membrane. Our lipidome analysis of GLTP knockout mice revealed alterations not only in HexCer but also in cholesterol, PE O-, and PS. Removing a single lipid often leads to changes across the entire lipid mixture in myelin, as lipids work together to maintain consistent biophysical properties like lipid order and fluidity[1]. How oligodendrocytes sense and adjust these parameters remains an important yet unresolved question.

In summary, we provide evidence for a role of tubular ER and nonvesicular lipid transfer in driving myelin growth around the axon. Our work raises a number of questions for future studies. Are there multiple lipid transfer proteins for the individual lipid classes? Are there other ER-to-myelin membrane nonvesicular lipid transport pathways involved in myelination?[27] If yes, how are they working together to coordinate their functions in generating membranes with the appropriate lipid mixture? Is the ER important for the recently reported calcium signaling in myelination?[64–66] How the ER shapes affects cellular physiology[37,67] and how ER membrane contact sites participate in lipid homeostasis are fundamental questions in biology. Oligodendrocytes are an excellent model system to begin to answer these questions, as they are highly specialized in generating huge amounts of lipids.

## Methods

### Generation of *Gltp* knockout (KO) and *Gltp* flox mice

*Gltp* KO and *Gltp* flox mice were generated by CRISPR/Cas9-assisted gene editing in mouse zygotes as described previously[68]. Briefly, pronuclear stage zygotes were obtained by mating C57BL/6 J males with superovulated C57BL/6 J females (Charles River). Embryos were then microinjected into the male pronucleus with a *Gltp*-specific CRISPR/Cas9 ribonucleoprotein (RNP) solution consisting of 30 ng/μl SpCas9 protein (IDT), 0.3 μM of each crRNA (IDT, "GGGGAGCTCTC TGCCGTGGTGGG" and "TGGCTGGCGGATGACTGTCAAGG"), 0.6 μM tracrRNA (IDT), and 10 ng/μl mutagenic long single-stranded DNA (lssDNA) lssGltp. LssDNA consisted of 200 nt homology arms and two loxP sites flanking exon 2 (ENSMUSE00000190151) of the *Gltp* gene (264 bp upstream and 487 bp downstream, respectively). After microinjection, treated zygotes were transferred into pseudopregnant CD-1 foster animals. To establish the KO and flox lines, mutant founder animals carrying either both loxP sites in cis or an 833 bp deletion (spanning exon 2) were identified and then crossed to C57BL/6 J animals to isolate the respective allele. Validation of the targeted loci was performed on genomic DNA from F1 animals by RFLP, Sanger sequencing, and qPCR-based copy number assays. To exclude additional unwanted modifications, putative off-target sites of the *Gltp*-specific crRNAs were predicted using the CRISPOR online tool[69]. Predicted loci were PCR-amplified and verified by Sanger sequencing. Validated animals without any off-target mutations were used for further breeding.

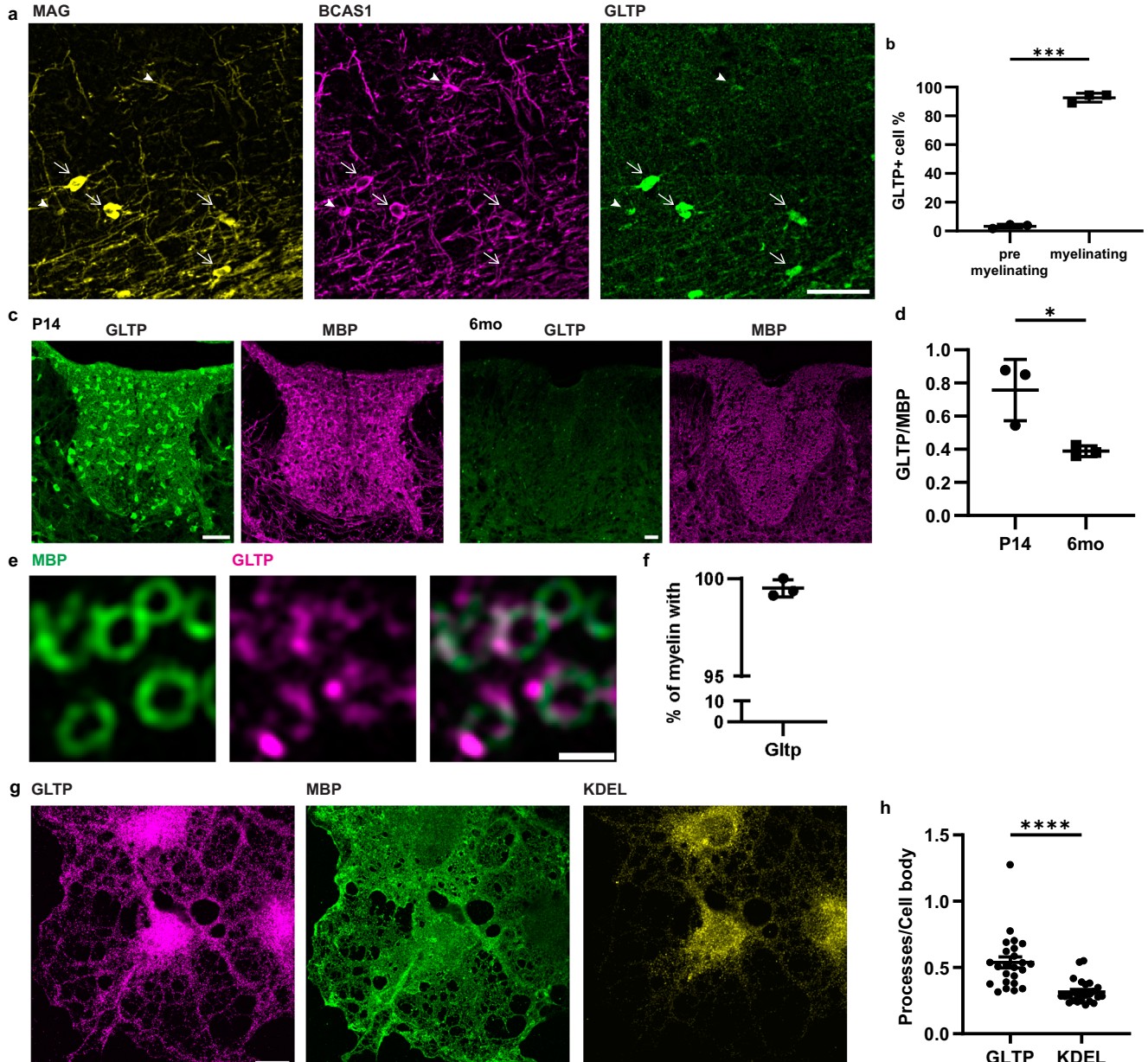

**Fig. 3 | Glycolipid transfer protein (GLTP) is associated with active myelination.**
**a** Immunohistochemistry of P14 mouse cortex. Arrow heads: pre-myelinating oligodendrocytes (MAG⁻BCAS1⁺); arrows: myelinating oligodendrocytes (MAG⁺BCAS1⁺). **b** Quantification of (**a**) showing mean ± SD, 3.33±1.36% pre-myelinating cells are GLTP⁺, 92.62 ± 3.08% myelinating cells are GLTP⁺ (*n* = 3 wild-type P14 mice, two-tailed paired t-test, t = 87.45, df=2, ***p = 0.0001) **c** Immunohistochemistry of P14 and six-month-old (6mo) adult mouse thoracic spinal cord, focusing on dorsal white matter. **d** quantification of GLTP fluorescence signal normalized by myelin (MBP) signal, in dorsal white matter of thoracic spinal cord, showing mean ± SD, P14: 0.76 ± 0.19, 6-month-old: 0.39 ± 0.03 (*n* = 3 wildtype mice for each time point, Two-tailed unpaired t-test, t = 3.403, df=4, *p = 0.0272). **e** zoom-in view of P14 spinal cord **f** Ratio of myelin (MBP⁺ ring) overlapping with GLTP. Quantification of (**e**) showing mean ± SD, 99.51 ± 0.44% myelin overlaps with GLTP (*n* = 3 wild-type P14 mice). **g** Immunocytochemistry of primary oligodendrocyte culture showing GLTP subcellular localization. **h** The ratio of fluorescence signals from GLTP and KDEL at the cellular processes to that at the cell body, showing mean ± SD, GLTP: 0.54 ± 0.20, KDEL: 0.32 ± 0.09 (*n* = 24 cells, two-tailed paired t-test, t = 5.834, df=23, ****p < 0.0001). Scale bars: 50 μm (**a**), 100 μm (**c**), 1 μm (**e**), 10 μm (**g**). Source data are provided as a Source Data file.

## Mouse experiments

All mice were handled according to institutional guidelines approved by the animal welfare and use committee of the government of Upper Bavaria (2532.Vet_ 02-21-67). All mice were group-housed in individually ventilated cages (IVC) in a specific pathogen-free and temperature-controlled (21 ± 2 °C) facility on a 12 h light/dark cycle, with *ad libitum* access to food and water. Both male and female mice were utilized but sex-matched among cohorts for comparison. All wild-type mice used in this study are C57BL/6j. The *Cnp-Cre* mouse line was a kind gift from Department Nave of Max Planck Institute for Multidisciplinary Sciences, and it is maintained as heterozygous by backcrossing with C57BL/6j. When crossing with the *Gltp flox* line, only female *Cnp-Cre* mice were used. All *Cnp-Cre* samples used in this study are heterozygous *Cnp-Cre*.

## Cell line and plasmid

U2OS cells (ATCC HTB-96) were cultured in DMEM medium with 10% FBS and 1X penicillin-streptomycin solution at 37 °C with 5% CO2.

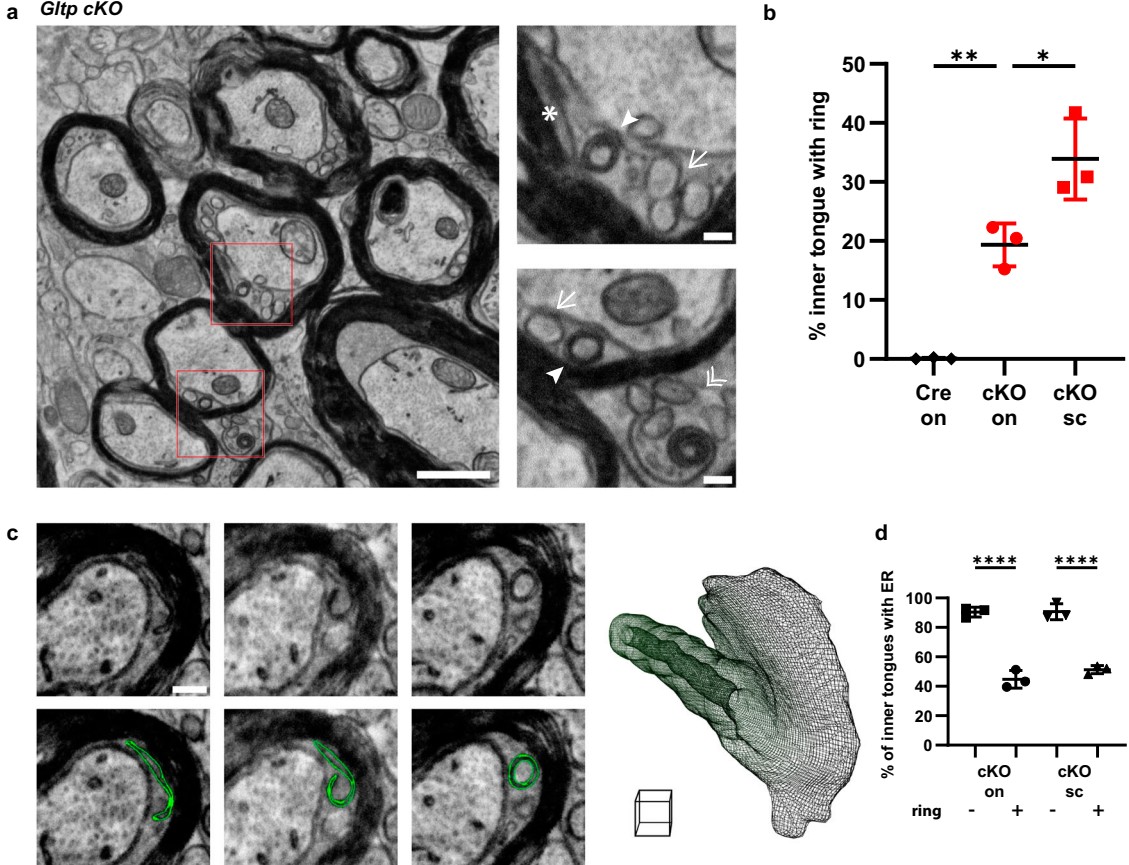

**Fig. 4 | Mice lacking GLTP in oligodendrocytes exhibit ER pathology in myelin.**
**a** Membrane rings in the inner tongue at the spinal cord of P14 *Gltp* cKO (*Cnp-Cre,*
*Gtlp^{flox/flox}*). Arrows: thin rings; arrowheads: thick rings; Star: a comet tail-like
structure associated with the ring; double arrowheads: rings in the outer tongue
**b** Ratio (mean ± SD) of inner tongue containing rings: 0.08 ± 0.13% of inner tongues
in the optic nerve of *Cnp-Cre* (Cre on), 19.34 ± 3.65% in the optic nerve of *Cnp-Cre,*
*Gtlp^{flox/flox}* (cKO on), 33.90 ± 6.87% in the spinal cord of *Cnp-Cre, Gtlp^{flox/flox}* (cKO sc)
(n = 3 mice for each condition, from 436 + 537 + 543 inner tongues of "Cre on",
508 + 525 + 531 of "cKO on", 589 + 585 + 664 of "cKO sc", One-way ANOVA: F (2,
6) = 42.77, Tukey's post-hoc test: Cre on vs. cKO on **p = 0.0046, cKO on vs. cKO sc

*p = 0.0173) **c** Volume EM of the optic nerve from *Gltp* cKO. Ring membrane is
segmented in green, and 3D reconstruction of ring from a 1.5 μm-thick stack
(31 slices with 50 nm interval), from dark green to light green across the stack. Scale
cube: 1 × 1 × 1 μm. **d** Amount of ER decrease in the presence of rings, suggested by
the ratio (mean ± SD) of inner tongue containing the ER: 90.34 ± 3.33% of ring- vs.
44.69 ± 6.01% of ring+ inner tongues in optic nerve of *Cnp-Cre, Gtlp^{flox/flox}* (cKO on),
90.67 ± 5.47% of ring- vs. 51.31 ± 2.78% of ring+ inner tongues in the spinal cord of
*Cnp-Cre, Gtlp^{flox/flox}* (cKO sc) (n = 3 mice for each condition, one-way ANOVA: F (3,
8) = 86.07, Sidak's multiple comparisons test: ****p < 0.0001) Scale bars: 1 μm (**a**),
150 nm (**a** zoom-in view) 200 nm (**c**). Source data are provided as a Source Data file.

Lipo3000 (Invitrogen) is used for transfection. The plasmid "pCMV-
UGT8-FusionRed" is constructed by Gibson assembly of the following
three fragments from 5′ to 3′: (1) pCMV from Addgene plasmid #36412;
(2) mouse Ugt8 cDNA (synthesized as gblock fragment by IDT); (3)
FusionRed from Addgene plasmid #54778. Because UGT8 is a Type I ER
membrane protein, we labelled it by fusing a non-cytotoxic fluores-
cence protein, FusionRed, at its C-terminal.

### Primary antibodies
rat anti-MBP (Abcam ab7349, 1:500 for IHC/ICC), chicken anti-MBP
(Thermo PA1-10008, 1:500 for IHC/ICC), anti-Rtn4 (Abcam ab47085,
1:500 for IHC/ICC), anti-Reep5 (Proteintech 14643-1-AP, 1:500 for
IHC/ICC), anti-BCAS1 (Santa Cruz, sc136342, 1:400 for IHC), anti-
MAG-Alexa647 (Santa Cruz sc-166849 AF647, 1:100 for IHC 1 h room
temperature), anti-Rtn1 (Sigma HPA044249, 1:250 for ICC), anti-
Sec61b (Sigma HPA049407, 1:400 for ICC), anti-KDEL (Enzo ADI-
SPA-827-D, 1:250 ICC), anti-GLTP (Sigma HPA056461, 1:300 for IHC/
ICC with antigen retrieval, 1:500 for Western Blot), anti-GalCer
(Merck MAB342, 1:1500 for ICC), anti-GalCer-Alexa488 (Merck
MAB342A4, 1:100 for ICC), anti-UGT8 (Proteintech 17982-1-AP, 1:500
for ICC), anti-CD140a (Biolegend 135902, 1:300 for primary culture
isolation).

### siRNA for oligodendrocyte culture
Cell permeable siRNAs were purchased from Horizon/Dharmacon, in
the "SMARTpool" format (a mixture of 4 siRNA): nontargeting (D-
001910-10-20), mouse Rtn4 (E-059578-00-0010), mouse Gltp (E-
058085-00-0010). 100x stock solution is prepared according to
manufacturer's instruction. The siRNA is added to the cells when
switching to a differentiation medium, with a final concentration of
1 μM, and incubated for 3 days.

### Transmission and scanning electron microscopy
Optic nerve and spinal cord tissue were dissected and immersion
fixed (P14) or perfusion fixed (P28) in 4% PFA (EM-grade, Science
Services), 2.5% glutaraldehyde (EM-grade, Science Services), 2 mM
calcium chloride in 1xPBS, pH 7.4 (Science Services) for 24 h. Short
(2–3 mm) pieces of the optic nerve and the 1 mm thick spinal cord
slices were incubated in the same fixative for another 24 h and
stored in 0.1 M cacodylate buffer at 4 °C. We applied a rOTO en bloc
staining protocol including postfixation in 2% osmium tetroxide
(EMS), 1.5% potassium ferricyanide (Sigma) in 0.1 M sodium caco-
dylate (Science Services) buffer (pH 7.4) (Kislinger et al.[70]). The
contrast was enhanced by incubation in 2% thiocarbohydrazide
(Sigma) for 45 min at 40 °C. The tissue was washed in water and

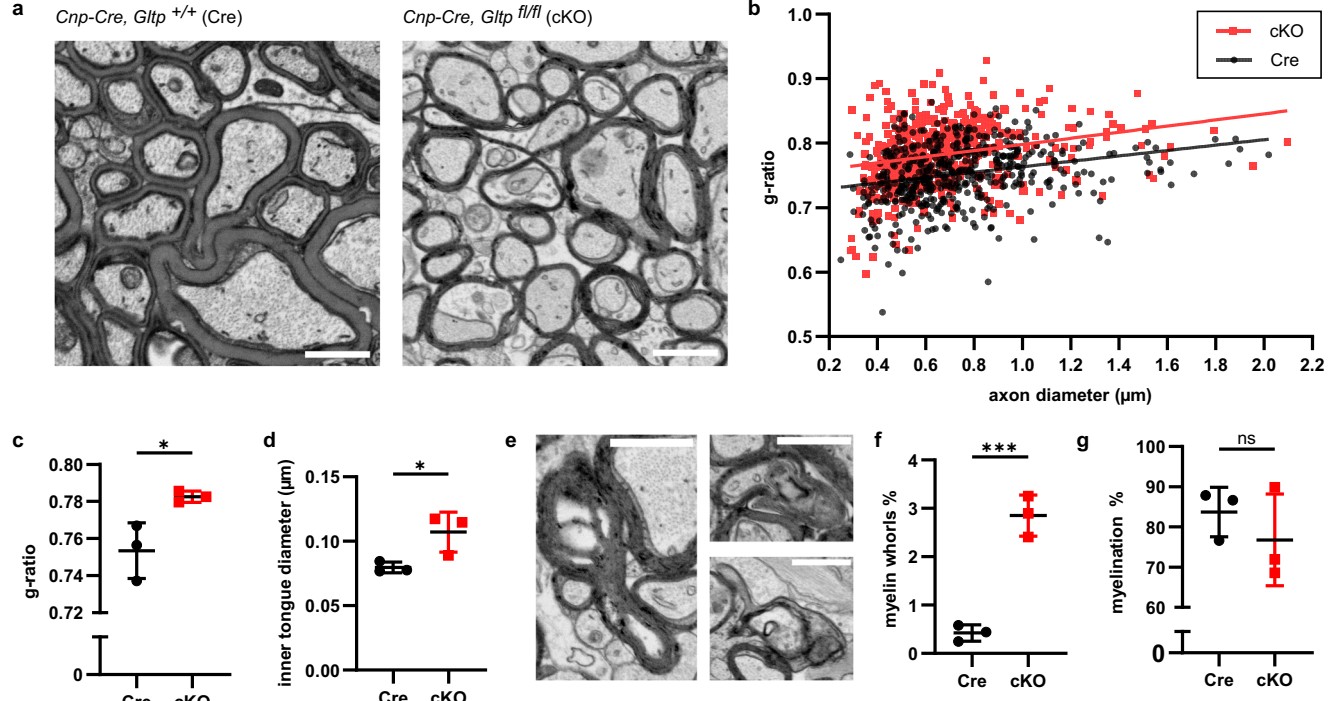

**Fig. 5 | Mice lacking GLTP in oligodendrocytes exhibit hypomyelination and dysmyelination. a** Optic nerve of *Gltp* cKO and Cre control mice at P28. **b, c** g-ratio (a measurement of myelin thickness defined as inner diameter of myelin divided by outer diameter), quantification from three mice for each condition. **b** shows g-ratio of individual myelin relative to the enwrapped axon's diameter. **c** shows the mean of myelin g-ratio from each mouse, showing mean ± SD, Cre: 0.754 ± 0.015, cKO: 0.783 ± 0.003 (*n* = 3 mice for each condition, Two-tailed unpaired t-test, t = 3.282, df=4, \**p* = 0.0304) **d** mean of inner tongue diameter from three mice for each condition, showing mean ± SD, Cre: 0.080 ± 0.004 µm, cKO: 0.107 ± 0.016 µm. (*n* = 3 mice for each condition, Two-tailed unpaired t-test, t = 2.943, df=4, \**p* = 0.0422) **e** examples of myelin whorls **f** Ratio of myelin whorls, showing mean ± SD, Cre: 0.424 ± 0.170%, cKO: 2.852 ± 0.425% (*n* = 3 mice per condition, Two-tailed unpaired t-test, *t* = 9.203, df=4, \*\*\**p* = 0.0008). **g** Percentage of axons that are myelinated, showing mean ± SD, Cre: 83.720 ± 6.148%, cKO: 76.780 ± 11.43% (*n* = 3 mice per condition, two-tailed unpaired t-test). Scale bars: 1 µm (**a** and **e**). Source data are provided as a Source Data file.

incubated in 1% aqueous osmium tetroxide, washed and further contrasted by overnight incubation in 1% aqueous uranyl acetate at 4 °C and 2 h at 50 °C. Samples were dehydrated in an ascending ethanol series and infiltrated with the epoxy-araldite resin LX112 (Ladd Research). Blocks were cured for 48 h at 60 °C, trimmed (EM TRIM, Leica) and sectioned at 100 nm thickness using a 35° ultra-diamond knife (Diatome) on an ultramicrotome (UC7, Leica). Sections were collected onto 1 × 0.5 cm carbon nanotube tape strips (Science Services) for scanning EM (SEM) analysis or onto formvar-coated copper grids (Plano) for transmission EM (TEM). For SEM imaging the samples on tape were attached to adhesive carbon tape (Science Services) on 4-inch silicon wafers (Siegert Wafer) and grounded by adhesive carbon tape strips (Science Services). EM micrographs were acquired on a Crossbeam Gemini 340 SEM (Zeiss) with a four-quadrant backscatter detector at 8 kV using ATLAS5 Array Tomography (Fibics). Medium lateral resolution images (100-200 nm) allowed the identification of regions of interest that were in turn reimaged at 4–10 nm lateral resolution. TEM micrographs for higher resolution representations were acquired on a JEM 1400plus (JEOL) equipped with a XF416 camera (TVIPS) and the EM-Menu software (TVIPS). Image analysis was performed in Fiji (Schindelin et al.,[71]).

## Volume electron microscopy

For volume EM analysis of myelinated axons in the optic nerve we applied Automated Tape Collecting Ultramicrotomy (ATUM)[72]. The blocks were trimmed at a 45° angle on four sides using a trimming machine (EM TRIM, Leica). Serial sections were taken with a 35° ultra-maxi diamond knife (Diatome) at a nominal cutting thickness of 50 nm on the ATUMtome (Powertome, RMC). Sections were collected onto

freshly plasma-treated carbon-coated Kapton tape (kindly provided by Jeff Lichtman and Richard Schalek). Plastic tape stripes were assembled onto adhesive carbon tape (Science Services) attached to 4-inch silicon wafers (Siegert Wafer) and grounded by adhesive carbon tape strips (Science Services). ATUM-SEM acquisition was performed on an Apreo S2 SEM (Thermo Fisher Scientific) with the T1 detector using the Maps2 (Thermo Fisher Scientific) software. The tissue was hierarchically imaged as previously described[70]. Target regions were identified and acquired at high resolution (4 × 4 × 50 nm). Serial section data were aligned by a sequence of automatic and manual processing steps in Fiji TrakEM2[71,73]. The VAST software was used for manual segmentation[74] and rendered in Blender[75] for three-dimensional visualization.

## EM analysis

G-ratio and inner tongue diameter were measured semi-automatedly using MyelTracer[76]. Outfolding and paranodes are excluded for quantification. Axon, inner and outer edge of myelin are recorded. The diameter of each compartment is calculated as $2*\sqrt{area/\pi}$. G-Ratio is calculated as $\sqrt{\frac{inner\ myelin\ area}{outer\ myelin\ area}}$, i.e. the ratio of inner to outer myelin diameter. Notably this is an updated definitions of g-ratio, as opposed to the ratio of axon diameter to outer myelin diameter, so that we can take into account of inner tongues present in developing myelin. Inner tongue diameter is calculated as inner myelin diameter - axon diameter.

ER membranes in ultrastructural data were identified according to morphological criteria and EM stain. The continuous network structure in the volume data sets distinguishes ER membranes from endocytotic or other vesicles[32,33]. While different staining protocols result in diverging grey values of organellar membranes, in several rOTO

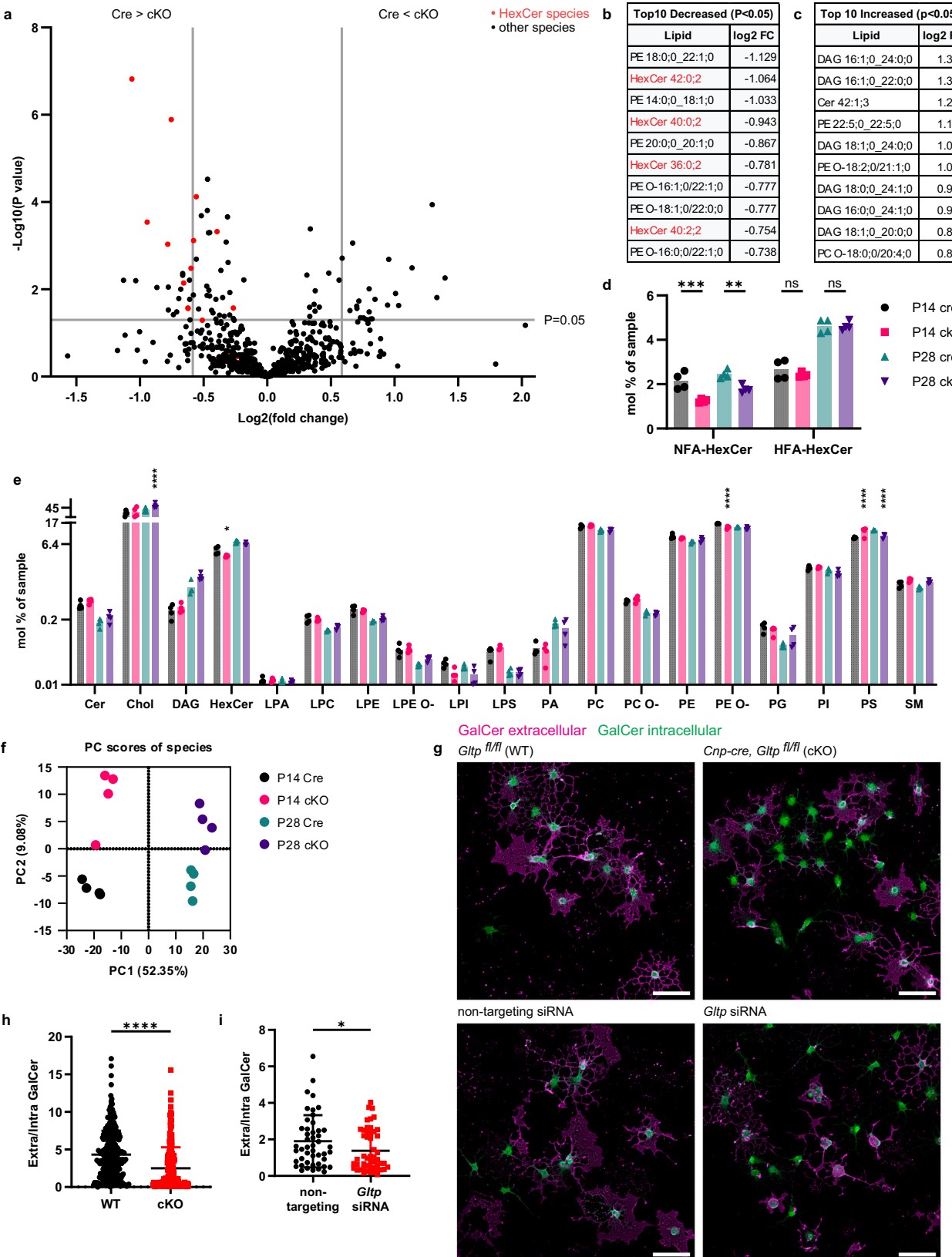

**b** Top10 Decreased (P<0.05)

| Lipid | log2 FC |
|---|---|
| PE 18:0;0_22:1;0 | -1.129 |
| HexCer 42:0;2 | -1.064 |
| PE 14:0;0_18:1;0 | -1.033 |
| HexCer 40:0;2 | -0.943 |
| PE 20:0;0_20:1;0 | -0.867 |
| HexCer 36:0;2 | -0.781 |
| PE O-16:1;0/22:1;0 | -0.777 |
| PE O-18:1;0/22:0;0 | -0.777 |
| HexCer 40:2;2 | -0.754 |
| PE O-16:0;0/22:1;0 | -0.738 |

**c** Top 10 Increased (p<0.05)

| Lipid | log2 FC |
|---|---|
| DAG 16:1;0_24:0;0 | 1.397 |
| DAG 16:1;0_22:0;0 | 1.332 |
| Cer 42:1;3 | 1.296 |
| PE 22:5;0_22:5;0 | 1.138 |
| DAG 18:1;0_24:0;0 | 1.033 |
| PE O-18:2;0/21:1;0 | 1.008 |
| DAG 18:0;0_24:1;0 | 0.955 |
| DAG 16:0;0_24:1;0 | 0.945 |
| DAG 18:1;0_20:0;0 | 0.846 |
| PC O-18:0;0/20:4;0 | 0.824 |

GalCer extracellular    GalCer intracellular

protocols ER membranes appear darker in comparison to other organellar membranes by TEM and SEM backscattered detection[77]. Additionally, in our datasets, the Golgi and endocytic compartments have lighter lumens compared to the ER. The ER-plasma membrane distance is measured as the shortest distance between the ER and the plasma membrane.

**Immunohistochemistry**

The mice were anesthetized with isoflurane, followed by transcardial perfusion with PBS and then 4% paraformaldehyde (PFA). Subsequently, brains and spinal cords underwent postfixation in 4% PFA for 4 h and overnight, respectively. The tissues were cryoprotected in 30% sucrose in PBS until sink. Tissue freezing on dry ice using Tissue-Tek

**Fig. 6 | Delivery of glycolipid to myelin is impaired in Gltp mutants. a–f** Lipidomics analysis of myelin purified from *Gltp* cKO (*Cnp-Cre Gltp^{flox/flox}*) and Cre control (*Cnp-Cre*) mice (*n* = 4 mice for each genotype/time point). Source data are provided as a Source Data file. **a** Volcano plot shows differentiated regulated lipid species in cKO vs. Cre control. Vertical lines mark -/+ 1.5 fold change. Data points above horizontal line have *p* value < 0.05. Data were grouped based on genotype (Cre and cKO), and transformed to log2 for two-tailed Welch's t-test. **b** and **c** Top 10 decreased and increased lipid species sorted by fold changes. P values are available in the source data. **d** Relative amount (mole %) of HexCer containing normal fatty acid (NFA) or 2-hydroxy fatty acid (HFA), mean ± SD: P14-Cre-NFA: 2.15 ± 0.39, P14-cKO-NFA: 1.25 ± 0.08, P28-Cre-NFA: 2.44 ± 0.19, P28-cKO-NFA: 1.78 ± 0.17, P14-Cre-HFA: 2.65 ± 0.44, P14-cKO-HFA: 2.42 ± 0.11, P28-Cre-HFA: 4.61 ± 0.3, P28-cKO-HFA: 4.62 ± 0.2. (Two-way ANOVA followed by Tukey's post-hoc tests, P14-Cre-NFA vs. P14-cKO-NFA ***p* = 0.0003, P28-Cre-NFA vs. P28-cKO-NFA ***p* = 0.0088). **e** Relative amount (mole %) of various lipid classes. Color code is the same as in **d**. (Two-way

ANOVA followed by Tukey's post-hoc test. Stars indicate significant changes compared to the control of same age. P28-Cre-Chol vs. P28-cKO-Chol ****p* < 0.0001, P14-Cre-HexCer vs. P14-cKO-HexCer **p* = 0.0278, P14-Cre-PE O- vs. P14-cKO-PE O- ****p* < 0.0001, P14-Cre-PS vs P14-cKO-PS ****p* < 0.0001, P28-Cre-PS vs P28-cKO-PS ****p* < 0.0001). **f** Principal Component Analysis (PCA) of lipid species from the samples. Each sample is myelin from one animal. **g** Mouse primary oligodendrocyte's extracellular(magenta) and intracellular(green) GalCer signal obtained before and after Triton X-100. **h** Quantification of WT and KO cells from three mice for each condition, showing mean ± SD, WT (*Gltp^{fl/fl}*): 4.32 ± 3.18, cKO (*Cnp-Cre, Gltp^{fl/fl}*): 2.50 ± 2.78. (*n* = 286 for WT, *n* = 265 for cKO, two-tailed unpaired t-test, t = 7.127, df=549, ****p* < 0.0001). P value is calculated by Student's t-test. **i** Quantification of non-targeting and *Gltp* siRNA, showing mean ± SD, non-targeting siRNA: 1.90 ± 1.43, *Gltp* siRNA: 1.38 ± 1.13. (*n* = 47 for nt, *n* = 50 for siRNA, Two-tailed unpaired t-test, t = 1.986, df=95, **p* = 0.0500) Scale bars: 50 μm **g**. Source data are provided as a Source Data file.

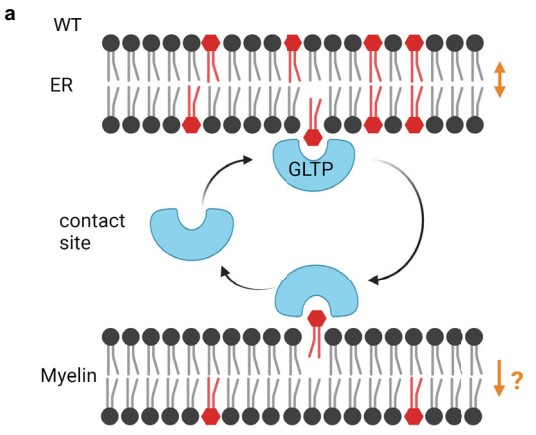
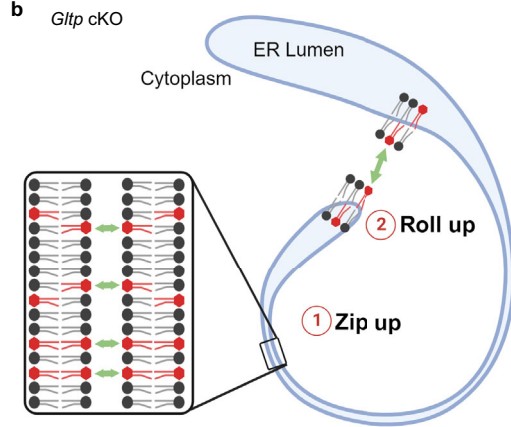

**Fig. 7 | Graphical illustration of model showing transport of GalCer in developing myelin. a** In wild-type myelin, GLTP transfers GalCer (red) from the ER to myelin membrane. Double-head arrow indicates GalCer reach an equilibrium in ER. Single-head arrow indicates that a yet-to-be-identified machinery transports GalCer to the extracellular leaflet of myelin membrane, keeping GalCer low in the intracellular leaflet of myelin membrane, creating a gradient and establishing the transfer direction as ER-to-myelin membrane. **b** Ring formation in *Gltp* cKO occurs in two steps. Step 1: GalCer attract each other and zip up the ER from the lumen. Step 2: GalCer accumulating in the cytoplasmic leaflet of ER make the ER to roll up. Created with BioRender.com.

O.C.T. was cut to 14-μm coronal sections using a Leica CM 1900 cryostat. Sections were rinsed with PBS containing and permeabilized in 0.3% Triton X-100 for 15 min (for brain) or 1 h (for the spinal cord). For GLTP staining, antigen retrieval was performed for 20 min sodium citrate buffer (pH 6) at 80 °C using water bath. Sections were incubated in blocking solution (1% FBS, 1% fish gelatin, and 1% bovine calf serum in PBS) for 1 h at room temperature and if mouse primary antibodies are used, Fab fragments are added for another 1 h. Primary antibodies, diluted in 10% blocking solution, were added and incubated overnight at 4 °C. On the following day, sections were incubated with secondary antibodies, diluted in 10% blocking solution, for 1 h at room temperature, followed by washing, staining with 2 μg/ml Hoechst 33342, and mounting with ProLong gold antifade mounting medium (Thermo P36934).

### Primary oligodendrocyte culture
Oligodendrocyte progenitor cells were prepared from P7-P9 C57BL/6 J mouse brains by immunopanning[78]. Briefly, cortices were dissociated to single-cell suspension, which was passed through two negative-selection plates coated with BSL1 to remove microglia. The remaining cell suspension was then incubated in a positive-selection plate coated with anti-CD140a antibodies. The attached cells were collected by accutase and cultured on poly-L-lysine-coated coverslips in proliferation medium containing Dulbecco's modified Eagle's medium (DMEM; Thermo Fisher Scientific, catalog no. 41965), Sato Supplement, B-27

Supplement, GlutaMAX, Trace Elements B, penicillin–streptomycin, sodium pyruvate, insulin, N-acetyl-L-cysteine, D-biotin, forskolin, ciliary neurotrophic factor (CNTF), platelet-derived growth factor (PDGF) and neurotrophin-3 (NT-3). The culture is differentiated by replacing the above-mentioned proliferation medium's PDGF and NT-3 with T3.

### Immunocytochemistry
Cells were fixed by 4% PFA in room temperature for 10 min, washed with PBS and stored in PBS containing azide at 4 °C. For staining, cells were permeabilized with PBS-0.1% Triton X-100 for 30 sec. For GLTP staining, antigen retrieval was performed for 10 min sodium citrate buffer (pH 6) at 70 °C using a water bath. Blocking solution (1% FBS, 1% fish gelatin, and 1% bovine calf serum in PBS) is applied for 30 min before incubating with the primary antibody in PBS-10% blocking solution at 4 °C overnight. The following day, the cells are stained with secondary antibdodies for 1 h at room temperature, followed by Hoechst 33342 for 10 min and mounted with ProLong glass antifade mounting medium (Thermo P36980). For GalCer measurement, cells are first incubated with blocking solution, stained with mouse anti-GalCer antibody with 10% blocking solution for 1 h at room temperature, then anti-mouse Alexa 555 for 1 h at room temperature, permeabilized with 0.1% Triton X-100 for 30 sec, blocked again, and stained overnight with chicken anti-MBP (or other primary antibody) in PBS-10% blocking solution at 4 °C overnight. The following day, the cells were stained with anti-chicken-Alexa 647 and anti-GalCer

conjugated with Alexa 488 for 1 h in room temperature, and lastly stained with Hoechst 33342(Sigma B2261) and mounted with mounting medium.

## Fluorescence imaging and analysis

All florescence images were acquired by the point scanning confocal microscope Zeiss LSM 900 equipped with Airyscan 2 module, and Plan-Apochromat 63×/1.2 oil immersion objective, Plan-Apochromat 40x/1,1water immersion objective, Plan-Apochromat 20x/0,8 objective, Plan-Neofluar 10x/0,3 objective. Images were analyzed using Fiji[71].

## Western blot analysis

Individual P14 mouse brains were homogenized using a sonicator. The resulting whole-brain lysates were loaded at 12 μg per lane onto a 12% TGX™ precast gel (BioRad #4561046). Following separation by SDS-PAGE, proteins were transferred onto a nitrocellulose membrane. The membrane was then blocked in 3% BSA in PBS-Triton for 30 min at room temperature, followed by overnight incubation with primary antibodies in PBS at 4 °C. After washing, the membrane was incubated with HRP-conjugated secondary antibodies in PBST for 1 h at room temperature. Subsequently, targeted proteins were detected using Pierce ECL substrate (Thermo Fisher #32106), and visualized using an Odyssey Fc imager from LI-COR.

## Myelin isolation

Myelin was isolated from mouse brain and spinal cord (P14) or mouse brain only (P28), using one animal for each sample. The protocol with two rounds of sucrose density centrifugation and osmotic shocks is as previously described[79], with some modifications. The ultracentrifugation was done using an SW41 Ti rotor. The tissues were homogenized with a sonicator in a solution containing 10 mM Hepes pH 7.4 and 0.32 M sucrose. The homogenized tissue was layered on 0.32/0.85 M sucrose gradient and centrifuged at 29,417 × $g$ for 35 min with low deceleration and acceleration. The crude myelin fraction was recovered from the interface, resuspended in ice-cold distilled water, and centrifuged at 29,417 × $g$ for 18 min. The hypo-osmotic shock was applied to the pellet two more times and pellets were collected at 2296 × $g$ for 18 min. The pellet from the last step was dissolved in Hepes buffer containing 0.32 M sucrose, then all the centrifugation steps and hypo-osmotic shocks were repeated one more round. Eventually, the purified myelin pellet was resuspended in 1 ml PBS and stored at −20 °C.

## Lipidomics

Each sample is myelin from one animal. Mass spectrometry-based lipid analysis was performed by Lipotype GmbH (Dresden, Germany) as described[80]. Lipids were extracted using a chloroform/methanol procedure[81]. Samples were spiked with internal lipid standard mixture containing: cardiolipin 14:0/14:0/14:0/14:0 (CL), ceramide 18:1;2/17:0 (Cer), diacylglycerol 17:0/17:0 (DAG), hexosylceramide 18:1;2/12:0 (HexCer), lyso-phosphatidate 17:0 (LPA), lyso-phosphatidylcholine 12:0 (LPC), lyso-phosphatidylethanolamine 17:1 (LPE), lyso-phosphatidylglycerol 17:1 (LPG), lyso-phosphatidylinositol 17:1 (LPI), lyso-phosphatidylserine 17:1 (LPS), phosphatidate 17:0/17:0 (PA), phosphatidylcholine 15:0/18:1 D7 (PC), phosphatidylethanolamine 17:0/17:0 (PE), phosphatidylglycerol 17:0/17:0 (PG), phosphatidyl-inositol 16:0/16:0 (PI), phosphatidylserine 17:0/17:0 (PS), cholesterol ester 16:0 D7 (CE), sphingomyelin 18:1;2/12:0;0 (SM), triacylglycerol 17:0/17:0/17:0 (TAG) and cholesterol D6 (Chol). After extraction, the organic phase was transferred to an infusion plate and dried in a speed vacuum concentrator. The dry extract was re-suspended in 7.5 mM ammonium formate in chloroform/methanol/propanol (1:2:4; V:V:V). All liquid handling steps were performed using Hamilton Robotics STARlet robotic platform with the Anti Droplet Control feature for organic solvents pipetting. Samples were analyzed by direct infusion on a QExactive mass spectrometer (Thermo Scientific) equipped with a

TriVersa NanoMate ion source (Advion Biosciences). Samples were analyzed in both positive and negative ion modes with a resolution of Rm/z = 200 = 280,000 for MS and Rm/z = 200 = 17,500 for MSMS experiments, in a single acquisition. MSMS was triggered by an inclusion list encompassing corresponding MS mass ranges scanned in 1 Da increments[82]. Both MS and MSMS data were combined to monitor CE, Chol, DAG and TAG ions as ammonium adducts; LPC, LPC O-, PC and PC O-, as formiate adducts; and CL, LPS, PA, PE, PE O-, PG, PI and PS as deprotonated anions. MS only was used to monitor LPA, LPE, LPE O-, LPG and LPI as deprotonated anions, and Cer, HexCer, and SM as formiate adducts. Data were analyzed with in-house developed lipid identification software based on LipidXplorer[83,84]. Data post-processing and normalization were performed using an in-house developed data management system. Only lipid identifications with a signal-to-noise ratio >5, and a signal intensity 5-fold higher than in corresponding blank samples were considered for further data analysis.

## Myelin proteomics analysis

Tryptic in solution digestion of the purified myelin fraction was performed according to the filter-aided sample preparation (FASP) protocol[46] followed by LC-MS-analysis. In brief, purified myelin fractions corresponding to 10 μg myelin protein were dissolved and lysed in lysis buffer (1% ASB-14, 7 M urea, 2 M thiourea. 10 mM DTT 0.1 M Tris pH 8.5). Homogenised samples were diluted with lysis buffer containing 2% CHAPS to reduce ASB-14 concentration below 0.1%. Subsequent steps were carried out in centrifugal filter units (30 kDa MWCO Vivacon® 500 spin filters, Sartorius). After depletion of detergents, protein samples were reduced by 10 mM dithiothreitol (DTT) and alkylated with iodoacetamide, followed by a buffer exchange (50 mM acetonitrile bicarbonate (ABC)/10% acetonitrile) in which the samples were enzymatically digested overnight at 37 °C with 400 ng trypsin (Promega). Peptides were recovered by an initial spin, followed by two rounds of centrifugation with 50 mM ABC. Subsequently, 1% trifluoroacetic acid (TFA) was added to the pooled flow-throughs before mixing peptides with a ratio 1:1 of 10% acetonitrile in 1% TFA. Peptides were desalted by custom-made C18 tips[85].

Samples were reconstituted in 0.1% (FA) and separated using an Easy nLC1200 liquid chromatograph (Thermo Scientific) followed by peptide detection on a Q Exactive HF mass spectrometer (Thermo Scientific). Separation was carried out on 75 μm × 15 cm custom-made fused silica capillary packed with C18AQ resin (Reprosil-PUR 120, 1.9 μm, Dr. Maisch) with a 140 min acetonitrile gradient in 0.1% FA at a flow rate of 400 nl/min (3–6 % ACN gradient for 2 min, 6–30% ACN gradient for 90 min, 30–44% ACN gradient for 20 min, 44–75% ACN gradient for 10 min, 75–100% ACN gradient for 5 min). Peptides were ionized using a Nanospray Flex Ion Source (Thermo Scientific). Peptides were identified in full MS / ddMS² (Top15) mode, dynamic exclusion was enabled for 20 s and identifications with an unassigned charge or charges of one or >8 were rejected. MS1 resolution was set to 120,000 with a scan range of 300–1700 m/z, MS2 to 15,000, and AGC target of 3e6. Raw data were analyzed using MaxQuant's (version 1.6.0.1)[86] Andromeda search engine in reversed decoy mode based on a Mus musculus reference proteome (Uniprot-FASTA, UP000000589, downloaded November 2023) with a false discovery rate (FDR) of 0.01 at both peptide and protein levels. Digestion parameters were set to specific digestion with trypsin with a maximum number of 2 missed cleavage sites and a minimum peptide length of 7. Oxidation of methionine and amino-terminal acetylation were set as variable and carbamidomethylation of cysteine as fixed modifications. The tolerance window was set to 20 ppm (first search) and to 4.5 ppm (main search). Label-free quantification was set to a minimum ratio count of 2, re-quantification and match-between-runs was selected and 4 biological replicates per condition were analyzed. The resulting raw output protein group files of MaxQuant were processed using Perseus

(version 2.0.7.0)[87]. In general, common contaminants, reverse and site-specific identifications were excluded and only proteins with a minimum of greater or equal 2 peptides (Peptides >= 2) were quantified. Only the proteins, which were identified and quantified in 3 out of 4 replicates for each condition were not excluded in the analysis. To obtain the relative difference in cKO vs WT, two-sample student's t-test was utilized and FDR or *p* value of 0.05 were set.

## Statistical analysis

Statistical Analysis were performed using Microsoft Excel 2016 and GraphPad Prism 10. Volcano Plot (Fig. 6a) additionally used Lipid Map statistical tool[88,89]. Missing values of lipidomics were filled as 0. For mouse analysis, mean value per mouse was calculated and presented as a single data point.

## Reporting summary

Further information on research design is available in the Nature Portfolio Reporting Summary linked to this article.

## Data availability

All other data that support the findings are available upon request from the authors. The lipidomics data generated in this study has been deposited in the Zenodo database with https://doi.org/10.5281/zenodo.13779035. The proteome data generated in this study has been deposited in PRIDE under accession code PXD055984 (http://www.ebi.ac.uk/pride/archive/projects/PXD055984; https://ftp.pride.ebi.ac.uk/pride/data/archive/2024/09/PXD055984). Source data are provided with this paper.

## Code availability

No code was generated in this paper.

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

## Acknowledgements

We thank Janos Groh for the discussion, thank Cornelia Niemann for technical assistance. The work was supported by grants from the German Research Foundation (TRR 128-2, Project ID 408885537-TRR 274, SyNergy Excellence Cluster, EXC2145, Project ID390857198), the Human Frontier Science Program (HFSP), the ERC (ADvG), and the Dr. Miriam and Sheldon G. Adelson Medical Research Foundation, Chan-Zuckerberg Initiative grant.

## Author contributions

J.W. and M.S. conceived the project and designed experiments. J.W., J.D., R.F., A.D.D., K.N., and C.B. performed or analyzed experiments. G.K. and M.Sch performed electron microscopy experiments. B.W. and W.W. generated transgenic mouse lines. J.W. and M.S. analyzed the results and wrote the manuscript.

## Funding

## Competing interests

The authors declare no competing interests.
