## [Transparent Peer Review file · Nature Communications]

Nonvesicular lipid transfer drives myelin growth in the central nervous system

Corresponding Author: Professor Mikael Simons

Version 0:

Reviewer comments:

Reviewer #1

(Remarks to the Author)

The article written by Wu et. al., aims to establish the significance of non-vesicular lipid transport from the Endoplasmic Reticulum (ER) to the myelin sheath in central nervous system myelination. The study addresses a critical gap in knowledge in the myelin field which is the mechanism by which lipids are continuously transported and integrated into the growing myelin membrane. The authors utilized a very versatile technique, Automated Tape-collecting Ultramicrotome-Scanning Electron Microscopy, to thoroughly visualize and reconstruct the ER in proximity to the myelin sheath. With this technique, they were able to show that the tubular ER was the most abundant organelle in the myelin inner tongue suggesting a potential role of the ER in myelination. Using immunofluorescence staining, they also showed that ER tubular proteins (Receptor Expression-Enhancing Protein 5 (REEP5) and Reticulon 4 (RTN4)) were enriched in actively myelinating oligodendrocytes compared to pre-myelinating oligodendrocytes. The role of tubular ER in myelination was supported by the reduction in oligodendrocyte cell size following Rtn4 knockdown. The rest of the study then pivots to identifying a lipid transfer protein that could facilitate the transfer of non-vesicular lipids from the ER to the myelin membrane. Glycolipid transfer protein (GLTP) was identified as a good candidate because of its enrichment in oligodendrocytes. They found that the conditional loss of GLTP in oligodendrocytes (Gltp cKO) leads to several deficits including hypomyelination, however, there was no difference in the number of myelinated axons. This suggests that GLTP may play a more crucial role in active myelination compared to the initiation of myelination. Given the abundance of galactosylceramide (GalCer) in myelin and its synthesis in the ER, the authors hypothesized that GLTP aids in the transfer of GalCer from the ER to promote myelination. To test this hypothesis, they performed lipidomic analysis on myelin isolated from WT and Gltp cKO mice. Unbiased analysis of the data revealed that HexCer (a combination of GalCer and glucosylceramide) was reduced in Gltp cKO mice. The reduced delivery of GalCer to myelin following GLTP loss was further supported by the lower extracellular to intracellular GalCer ratio in oligodendrocytes purified from Gltp cKO. Taken together, this study enhances our fundamental understanding of the myelination process. The findings have revealed the intricate nature of the ER in the myelin inner tongue and uncovered a novel function of GLTP in directly transferring GalCer lipids from the ER to the growing myelin sheath to promote active myelination.

The strengths of this study were the logical flow of experiments from the characterization of tubular ER in myelin to the utilization of a mouse model to show the role of ER non-vesicular lipid transfer in myelination. The experiments utilized also directly answered the questions asked by the authors and the interpretations of the results were reasonable.

Despite the rigor and novelty of the study, several aspects could be improved based on the comments below. Additional information and studies would need to be implemented to further connect the ER-to-myelin membrane nonvesicular lipid transport pathway.

Major points:

1. The authors utilized two markers of the tubular ER, (REEP5 and RTN4). To further substantiate the presence of the ER in myelin, complementary markers are needed.
2. The use of the primary oligodendrocyte cell culture system was a good tool to visualize the ER, however, it wasn't sufficient to conclude how the loss of the tubular ER protein Rtn4 affects myelination. The only readout was the reduced oligodendrocyte cell size which doesn't necessarily indicate impaired myelination. A more direct readout of the effect of Rtn4 loss on myelination could be incorporated. This could include MBP staining of the primary oligodendrocytes or potentially

hypomyelination in an Rtn4 deficient mouse model. Together, these will strengthen the conclusion that the loss of tubular ER protein Rtn4 impairs myelination.

3. The rationale for focusing on GLTP as a mechanism for the transfer of non-vesicular lipids was not very strong. Other lipid transfer proteins in the cytoplasm could have the same function, therefore, a supplementary figure needs to be included that highlights the relative expression of GLTP in oligodendrocytes compared to other non-vesicular lipid transfer proteins.

4. No comments were made on the observation of the fluctuations in other important myelin lipid components such as cholesterol and phosphatidylserine. The changes in these lipids following GLTP deletion could indicate that GLTP can also regulate myelination independent of GalCer transfer.

5. While the lipidomic analysis provided direct evidence for the importance of GalCer in myelination, there could be other effects of GLTP deletion. In this case, it would be beneficial to add a layer of transcriptomic profiling to understand the gene expression changes that are occurring with GLTP deletion.

6. GLTP may have other functions in addition to lipid transfer, for example, forming functional complexes with other proteins. Specifically mutating the functional domain of GLTP that is important for binding and transferring GalCer would allow the authors to attribute its transport function to myelination. This experiment would also eliminate the possibility that GLTP's other functions contribute to myelination.

7. Considering that GLTP cKO maintains compact myelin, only inducing hypomyelination but no further myelin deficiencies described. While it helps understand the process of myelin growth, it doesn't seem like a critical protein necessary for myelination. It would be extremely helpful for the translational impact of this study to clarify how this affects mice as they age. Is there a change in the mouse cKO motor phenotype or life expectancy? Are there related GLTP deficient diseases?

8. The long discussion seems to bring up more questions than summarize the actual innovative investigations done in this study. Unfortunately, because of this, it paints the results as short and underwhelming since there aren't further studies past the lipidomic results, ending abruptly.

Minor points:

1. Provide more markers for tubular ER presence in myelin. The two markers used (REEP5 and RTN4) isn't sufficient to validate it.
2. The oligodendrocyte size reduction in Rtn4 loss doesn't correlate in impaired myelination. Further myelin studies would need to be conducted in order to support that claim.
3. Figure 1e and figure 3e would be visualized better by keeping the color of the staining instead of black and white.
4. Figure 2h graph names seemed very close together and can be spaced out for better deciphering.
5. Typo in title of figure 6, "Delievery".
6. Maintain consistency between "min" or "minutes", "h" or "hours" in the methods section.

Reviewer #2

(Remarks to the Author)

Wu et al. conducted an interesting study that reveals, for the first time, the presence of tubular endoplasmic reticulum (ER) in the inner tongue of the myelin sheath. Using cut-edge volume electron microscope technology, they discovered tubular ER network is in the myelin inner tongue and close to the myelin sheath. To test the hypothesis that direct lipid transfer between ER and plasma membrane contribute to myelin growth, they conditionally deleted GLTP, an oligodendrocyte enriched lipid transfer protein, in developing oligodendrocytes. They found GLTP cKO leads to ER pathology, a decrease in myelin glycolipid content, and hypomyelination.

This manuscript is well-written and easy to follow. Their figures are well organized. Their discoveries are novel and are supported by a series of well-designed experiments, for the most part. However, there are major concerns listed below that preclude the publication of this work, at least in its current form.

The biggest weakness for this paper is that GLTP deletion affects more than lipid transfer between tubular ER and myelin sheath. GLTP is a multi-functional protein, which mediates lipid transfer among intracellular membranes, not just ER to plasma membrane. A recent study suggests that GLTP is also involved in vesicle transport to plasma membrane (PMID: 36924944). This raised the question whether the observed phenotypes after GLTP deletion come from defects in direct lipid transfer between ER to myelin or from deficits in indirect lipid transfer from intracellular vesicles to myelin. Evidence for a role of non-vesicular lipid transfer in myelination is lacking.

In addition to this major weaknesses, below are some additional suggestions that may help to improve this study.

1. Tubular ER is present in most of the animal cells. The existence of tubular ER in myelin sheath suggests tubular ER can support myelin growth, but it does not explain why oligodendrocytes, not other brain cell types, can produce enormous amount of lipids. One possibility is ER makes more contact with plasma membrane in oligodendrocytes. A quantification of tubular ER-myelin sheath coverage will help to clarify this issue. Also, a comparison of several tubular ER lipid transport gene expressions among different brain cell types will tell us if oligodendrocytes have more ER lipid transport machinery. Please also clarify whether tubular ER in inner tongue is still connected with ER network in the cell body.
2. Page 6, line 1: The authors observed a decrease in cell size after knocking down Rtn4 and concluded that tubular ER is crucial in myelination. It is more appropriate to use "membrane addition" instead of "myelination" as there is no myelination in their in vitro model.
3. Page 8 line 1 and figure 5: They found hypomyelination and axon degeneration in GltP knockout mice. As axon degeneration and demyelination often happen together, whether the observed "hypomyelination" phenotype is due to developmental hypomyelination or due to demyelination is not clear. Please clarify this. Also, please quantify the ratio of dysmyelination (abnormal myelination pattern) if there is any.
4. Page 14, figure 1g: Are the ERs in the inner tongue closer to the inner tongue plasma membrane facing the axon or the

inner tongue plasma membrane facing outer layers of myelin? Do the author envision lipid transfer to occur on the side facing the axon or the side facing outer myelin layers? Please also include how they define the distance between myelin and tubular ER in the method section. Is it the closest distance between tubular ER and tubular?

5. Page 16, figure 2: they found tubular ER markers, REEP5 and RTN4 are dramatically reduced in 6 months old mice as compared to P14. Are there fewer tubular ER in the myelin sheath of 6 months old mice as well.
6. Page 17, figure 3: the intensities of GLTP seems stronger in the cell body than in branches. Please provide the quantification of cell body/process ratio of GLTP as they did in figure 2h. This information could be helpful for understanding whether GLTP mostly functions in the cell body or processes. Also, does Gltp knockout affect ER morphology in the cell body?
7. Page 18, figure 4: In figure 4a and 4c, although ER morphology is changed in Gltp knockout mice, it is unclear whether the distance between ring ER and myelin plasma membrane is affected. A quantification would be informative.
8. Page 20, figure 6g: from the image they provided, it looks like there are more star/arborized shaped cells in Gltp KO group. This may suggest that Gltp KO also affects the maturation of oligodendrocytes. Please quantify the number of oligodendrocytes at different maturation stages based on their morphology.
9. Besides citing several references, it is unclear how ER was identified and confirmed on EM. How can the authors be certain that those membraneous structures are ER and not Golgi, endosomes, lysosomes, or other compartments?
10. Another suggestion is to perform live imaging to show the dynamics of ER and plasma membrane contact events in oligodendrocytes. FRET assays can be considered

Reviewer #3

(Remarks to the Author)

In this study, the authors use volume electron microscopy to visualize the ER during myelination. The authors demonstrate that the tubular ER is present in the inner layer of myelin where it is growing. They also explore the role of glycolipid transfer protein (GLTP) in myelination with the use of a conditional Gltp knockout mouse strain. They show that loss of Gltp leads to defects in the ER, myelin formation, and levels of HexCer – a major myelin lipid. These findings are also validated in primary OPC cultures.

The study is very well written and the conclusions are well justified by the data shown. The findings reveal important new insights into how lipids are installed into newly forming myelin membranes and the roles of the ER and GLTP in this process.

Some minor concerns and questions are noted below.

- p. 3 – How are the ER organelles definitively distinguished from other types of organelles?
 - p. 4 – In results section, should specify that the “white matter” in Fig. 2d/f is from spinal cord
 - p. 6 – could specify what cells CNP-Cre targets
 - p. 8 – From reading the methods sections, the lipidomics methodology is a targeted approach – not an untargeted approach. Not distinguishing between GalCer and GlcCer does not make it untargeted.
 - p. 8 - are lipids really “downregulated and upregulated” in the same way as genes and proteins? – I would argue that this should just be “increased and decreased”
 - p. 8 – how is the MS method distinguishing the individual FA species and why are they not identified for the HexCer species? I found the answer in the referenced methods paper, but it should perhaps be included here as well.
 - p. 8 – Is the PCA (Fig. 6f) performed on the individual lipids in Fig. 6a-c or on the total levels shown in Fig. 6e?
 - p. 8 – All of the lipid data is reported as mol%, which is a normal/standard way of reporting these values. Is this normalized by total moles of lipid measured by mass spec? Or by total lipid extract weight? Are there differences in the total lipid levels between the cKO and cre-only? Have you normalized by protein amount or tissue weight? This could provide a different view of the dataset.
- Fig. 6 – the caption mentions that the raw lipidomics data is in the supp files, but I did not see it.

Version 1:

Reviewer comments:

Reviewer #1

(Remarks to the Author)

I am pleased to see the revised manuscript and the adjustments that have been made for the story to be more cohesive. I deeply appreciate the meticulous actions taken by the authors in addressing all reviewer suggestions and points. The authors justified the use of REEP5 and RTN4 as these are the most abundantly expressed tubular ER markers in oligodendrocytes. The authors addressed my points further, especially with the inclusion of new tubular ER marker expression, IHC/volume EM of other reticulons, and quantification of the MBP sheet in primary oligodendrocyte cultures after tubular ER protein knockdown, which supports their claim that tubular ER is present and effects myelination. The author's

rationale was strengthened with the addition of gene expression profiles in nonvesicular lipid transport, which demonstrates GLTP as the most abundant lipid transfer protein in oligodendrocytes, supporting GLTP as a strong target mechanism for the transfer of non-vesicular lipids. It's also greatly appreciated for the distinction of the term dysmyelination instead of degeneration, as well as the clear description given for the identification of ER in EM. The additional layer of transcriptomic profiling analysis and quantification of dysmyelination profiles under GLTP deletion greatly contributed to understanding the large impact GLTP deficiency has on myelination and showed the upregulation of neurodegeneration and other supplemental pathways. The authors have also included a section in the discussion to address potential compensatory mechanisms in response to the fluctuations of other myelin lipid components under GLTP deletion. Lastly, mentioning the non-canonical functions of GLTP improve the discussion and give a well-rounded view on the topic for the audience. The revision has improved immensely with all actions taken from the authors, and I believe these changes significantly improved the manuscript and is an overall interesting read.

Reviewer #2

(Remarks to the Author)

The authors have addressed all my concerns. I recommend the publication of this manuscript.

Reviewer #3

(Remarks to the Author)

The authors have appropriately addressed all of the concerns raised in the initial review. I would recommend that this important study be published.

Response to reviewers:

General remarks:

We would like to thank the reviewers for their constructive and helpful comments. As you will see below, we have responded to all the issues raised and addressed them with new experiments and analyses, as far as possible. Our responses to each of the reviewer's comments are highlighted in blue below. We believe that the manuscript has been significantly improved following the recommendations of the reviewers, and we hope that the work is now considered acceptable for publication in *Nature Communication*.

Reviewer #1 (Remarks to the Author):

The article written by Wu et. al., aims to establish the significance of non-vesicular lipid transport from the Endoplasmic Reticulum (ER) to the myelin sheath in central nervous system myelination. The study addresses a critical gap in knowledge in the myelin field which is the mechanism by which lipids are continuously transported and integrated into the growing myelin membrane. The authors utilized a very versatile technique, Automated Tape-collecting Ultramicrotome-Scanning Electron Microscopy, to thoroughly visualize and reconstruct the ER in proximity to the myelin sheath. With this technique, they were able to show that the tubular ER was the most abundant organelle in the myelin inner tongue suggesting a potential role of the ER in myelination. Using immunofluorescence staining, they also showed that ER tubular proteins (Receptor Expression-Enhancing Protein 5 (REEP5) and Reticulon 4 (RTN4)) were enriched in actively myelinating oligodendrocytes compared to pre-myelinating oligodendrocytes. The role of tubular ER in myelination was supported by the reduction in oligodendrocyte cell size following Rtn4 knockdown. The rest of the study then pivots to identifying a lipid transfer protein that could facilitate the transfer of non-vesicular lipids from the ER to the myelin membrane. Glycolipid transfer protein (GLTP) was identified as a good candidate because of its enrichment in oligodendrocytes. They found that the conditional loss of GLTP in oligodendrocytes (GltP cKO) leads to several deficits including hypomyelination, however, there was no difference in the number of myelinated axons. This suggests that GLTP may play a more crucial role in active myelination compared to the initiation of myelination. Given the abundance of galactosylceramide (GalCer) in myelin and its synthesis in the ER, the authors hypothesized that GLTP aids in the transfer of GalCer from the ER to promote myelination. To test this hypothesis, they performed lipidomic analysis on myelin isolated from WT and GltP cKO mice. Unbiased analysis of the data revealed that HexCer (a combination of GalCer and glucosylceramide) was reduced in GltP cKO mice. The reduced delivery of GalCer to myelin following GLTP loss was further supported by the lower extracellular to intracellular GalCer ratio in oligodendrocytes purified from GltP cKO. Taken together, this study enhances our fundamental understanding of the myelination process. The findings have revealed the intricate nature of the ER in the myelin inner tongue and uncovered a novel function of GLTP in directly transferring GalCer lipids from the ER to the growing myelin sheath to promote active myelination.

The strengths of this study were the logical flow of experiments from the characterization of tubular ER in myelin to the utilization of a mouse model to show the role of ER non-vesicular lipid transfer in myelination. The experiments utilized also directly answered the questions asked by the authors and the interpretations of the results were reasonable.

Despite the rigor and novelty of the study, several aspects could be improved based on the comments below. Additional information and studies would need to be implemented to further connect the ER-to-myelin membrane nonvesicular lipid transport pathway.

We thank the reviewer for the supportive comments and helpful suggestions made.

Major points:

1. The authors utilized two markers of the tubular ER, (REEP5 and RTN4). To further substantiate the presence of the ER in myelin, complementary markers are needed.

We utilized REEP5 and RTN4 as markers for tubular ER in oligodendrocytes, because these two proteins are expressed at particular high levels in oligodendrocytes. Other reticulons such as Rtn2, Reep1, Reep2, Reep4 and Reep 6 are expressed at very low levels in oligodendrocytes (see Figure below). To extend our analysis to other reticulons, we chose Reticulon 1 (RTN1), which is abundantly expressed in oligodendrocytes, but also in neurons. When we performed immunohistochemistry on spinal cord cross-sections from P14 mice. RTN1 appeared as puncta on MBP⁺ rings, and quantification from three mice revealed that 81% have reticulon 1 puncta, similar to the observations for REEP5 and RTN4. (See new Supplementary Figure 1e, and Figure below). We also tried using an antibody against Reticulon 3, but the immunohistochemistry signal was very weak in the entire brain (data not shown). Thus, by combining IHC and volume EM, we are confident about our conclusion that tubular ER is present in developing myelin.

2. The use of the primary oligodendrocyte cell culture system was a good tool to visualize the ER, however, it wasn't sufficient to conclude how the loss of the tubular ER protein Rtn4 affects myelination. The only readout was the reduced oligodendrocyte cell size, which doesn't necessarily indicate impaired myelination. A more direct readout of the effect of Rtn4 loss on myelination could be incorporated. This could include MBP staining of the primary oligodendrocytes or potentially hypomyelination in an Rtn4 deficient mouse model. Together, these will strengthen the conclusion that the loss of tubular ER protein Rtn4 impairs myelination.

As suggested by the reviewer, we have now quantified MBP positive sheet area in primary cultures of oligodendrocytes after Rtn4 knockdown. As shown below, we found a reduction in MBP⁺ sheet size. The data has been added to Supplementary Figure 2g (see also below).

3. The rationale for focusing on GLTP as a mechanism for the transfer of non-vesicular lipids was not very strong. Other lipid transfer proteins in the cytoplasm could have the same function, therefore, a supplementary figure needs to be included that highlights the relative expression of GLTP in oligodendrocytes compared to other non-vesicular lipid transfer proteins.

Thank you for this comment. We apologize for not explaining why we decided to focus on GLTP. We have now added an analysis showing the expression level of the known lipid transfer proteins (based on the RNA-Seq dataset of Zhang et al.). GLTP stands out as the most highly expressed lipid transfer protein in oligodendrocytes. NPC1 and 2 are also expressed at relatively high levels, but these proteins operate in lysosomes. The analysis has been added to Supplementary Figure 3a (see also below).

Gene Expression Profiles Involved in Nonvesicular Lipid Transport (RNA-seq from Zhang et al. 2014)

4. No comments were made on the observation of the fluctuations in other important myelin lipid components such as cholesterol and phosphatidylserine. The changes in these lipids following GLTP deletion could indicate that GLTP can also regulate myelination independent of GalCer transfer.

Thank you for raising this point. We have now included a brief section in the discussion addressing why other lipids may also undergo changes in myelin: A notable aspect of myelin lipid composition is the presence of compensatory mechanisms. The removal of one lipid results in alterations throughout the entire lipid mixture in myelin. Lipids function collectively, maintaining constant biophysical parameters such as lipid order and fluidity. For instance, the deletion of the enzyme necessary for producing galactosylceramide causes an increase in glucosylceramide levels. How oligodendrocytes sense and adjust these parameters is an important yet unresolved question.

5. While the lipidomic analysis provided direct evidence for the importance of GalCer in myelination, there could be other effects of GLTP deletion. In this case, it would be beneficial to add a layer of transcriptomic profiling to understand the gene expression changes that are occurring with GLTP deletion.

We have now added another layer of analysis, by performing proteomics analysis of myelin purified from P28 *Glt1* cKO and Cre control mice. We decided to perform proteomics on myelin to provide a comparison to the lipidomics data, which was also performed on purified myelin. The quality of the samples was validated by principal component analysis, cellular component GO term enrichment analysis and *Glt1* reads. Enrichment analysis indicated upregulation of pathways such as neurodegeneration and lipid droplet organization in *Glt1* cKO (Fig. 7f), consistent with our electron microscopy and lipidomics findings. Two lipid transfer proteins were upregulated in cKO: *Mospd2* and *C2cd2l*. The data is shown in a new Supplementary Figure 5c-h (see also below).

Supplementary Fig. 9 | Myelin proteome alterations of *GltP* cKO

a Principal Component Analysis of the samples. **b** Cellular component GO term analysis showing these purified myelin samples contains mainly myelin, with some contaminations from axons and mitochondria. **c** Volcano plot shows differentiated regulated proteins in cKO compared to Cre control. Vertical lines mark ± 2 folds change. Data points above horizontal line have p -value < 0.05 . Two lipid transfer proteins were upregulated in cKO: *Mospd2* and *C2cd2l*. **d** Bar plot showing fold changes of transmembrane proteins among Top 200 abundant proteins. Note that mitochondria transmembrane proteins are not included because

they are known contaminant of biochemical purifications of myelin. e Bar plot showing fold changes of cytosolic proteins among Top 200 abundant proteins. f Upregulated pathways in *Gltp* cKO based on Metascape enrichment analysis

6. GLTP may have other functions in addition to lipid transfer, for example, forming functional complexes with other proteins. Specifically mutating the functional domain of GLTP that is important for binding and transferring GalCer would allow the authors to attribute its transport function to myelination. This experiment would also eliminate the possibility that GLTP's other functions contribute to myelination.

Reviewer 2 raised a similar issue, noting that a previous publication indicated GLTP may also play a role in vesicular trafficking. GLTP is a well-established lipid transfer protein, as consistently demonstrated in various previous studies and proven through in vitro lipid transfer assays. However, it is possible that GLTP has non-canonical functions beyond lipid transfer. Additionally, GLTP deletion might indirectly affect vesicular trafficking, for example, by disrupting ER morphology and function. Our proteome and lipid analyses enabled us to test for disruptions in vesicular trafficking, which would manifest as a depletion of membrane proteins from myelin. However, most transmembrane proteins did not show significant changes, and overall, there was no pattern of decrease, contradicting the hypothesis, that GLTP directly affects vesicular trafficking (see Supplementary Fig5 e-g). Instead, our lipid analysis revealed that HexCer constitutes the majority of reduced lipids. Thus, we conclude that GLTP influences lipid trafficking to myelin. We cannot exclude other non-canonical functions of this protein, and now mention this in the discussion. The reviewer suggested an interesting experiment: rescuing the GLTP KO with a wild-type GLTP lacking the lipid transfer domain. While this is an excellent suggestion, it is beyond the scope of this revision, as it would require generating transgenic mice with knock-in constructs at the endogenous locus, a process that would take at least two years. Moreover, performing rescue through viral vectors presents challenges because it leads to non-physiological levels, which is problematic because precise stoichiometry is often required for molecular complexes involving lipid transfer proteins.

7. Considering that GLTP cKO maintains compact myelin, only inducing hypomyelination but no further myelin deficiencies described. While it helps understand the process of myelin growth, it doesn't seem like a critical protein necessary for myelination. It would be extremely helpful for the translational impact of this study to clarify how this effects mice as they age. Is there a change in the mouse cKO motor phenotype or life expectancy? Are there related GLTP deficient diseases?

We agree with the reviewer that it is currently unclear whether GLTP has any disease implications. To date, no mutations have been identified that are associated with myelin disorders. It is important to note that the deletion of lipid transfer proteins generally has no impact on lipid transport in most cells, as there is significant compensatory activity (see for example:lipid transfer proteins E-syts triple knockout mice lack major phenotypes PMID: 27348751; 28301744). Remarkably, in our study, we demonstrate that the deletion of GLTP in oligodendrocytes leads to a distinct phenotype with ER pathology, hypomyelination and less HexCer in myelin. To determine whether GLTP knockout contributes to degenerative phenotypes, we quantified myelin dysmyelination profiles in WT and GLTP KO mice. We observed a significant increase in myelin alterations in GLTP KO mice, further indicating that its function cannot be fully compensated. The data is presented in a new Supplementary Figure 5j (see below).

8. The long discussion seems to bring up more questions than summarize the actual innovative investigations done in this study. Unfortunately, because of this, it paints the results as short and underwhelming since there aren't further studies past the lipidomic results, ending abruptly.

We agree with the reviewer that the discussion brings up many new question. We believe it is important to raise this question and to suggest possible strategies for addressing them, to provide a framework for future research.

Minor points:

1. Provide more markers for tubular ER presence in myelin. The two markers used (REEP5 and RTN4) isn't sufficient to validate it.

See above, we have now added RTN1 as an additional marker.

2. The oligodendrocyte size reduction in Rtn4 loss doesn't correlate in impaired myelination. Further myelin studies would need to be conducted in order to support that claim.

See above (point 2).

3. Figure 1e and figure 3e would be visualized better by keeping the color of the staining instead of black and white.

We now changed, as suggested.

4. Figure 2h graph names seemed very close together and can be spaced out for better deciphering.

We corrected

5. Typo in title of figure 6, "Delievery".

We corrected

6. Maintain consistency between “min” or “minutes”, “h” or “hours” in the methods section.

We corrected

Reviewer #2 (Remarks to the Author):

Wu et al. conducted an interesting study that reveals, for the first time, the presence of tubular endoplasmic reticulum (ER) in the inner tongue of the myelin sheath. Using cut-edge volume electron microscope technology, they discovered tubular ER network is in the myelin inner tongue and close to the myelin sheath. To test the hypothesis that direct lipid transfer between ER and plasma membrane contribute to myelin growth, they conditionally deleted GLTP, an oligodendrocyte enriched lipid transfer protein, in developing oligodendrocytes. They found GLTP cKO leads to ER pathology, a decrease in myelin glycolipid content, and hypomyelination.

This manuscript is well-written and easy to follow. Their figures are well organized. Their discoveries are novel and are supported by a series of well-designed experiments, for the most part. However, there are major concerns listed below that preclude the publication of this work, at least in its current form.

The biggest weakness for this paper is that GLTP deletion affects more than lipid transfer between tubular ER and myelin sheath. GLTP is a multi-functional protein, which mediates lipid transfer among intracellular membranes, not just ER to plasma membrane. A recent study suggests that GLTP is also involved in vesicle transport to plasma membrane (PMID: 36924944). This raised the question whether the observed phenotypes after GLTP deletion come from defects in direct lipid transfer between ER to myelin or from deficits in indirect lipid transfer from intracellular vesicles to myelin. Evidence for a role of non-vesicular lipid transfer in myelination is lacking.

We thank the reviewer for the supportive comments and helpful suggestions made.

The reviewer refers to a previous publication indicated GLTP may also play a role in vesicular trafficking. GLTP is a well-established lipid transfer protein, as consistently demonstrated in various previous studies and proven through in vitro lipid transfer assays (PMID: 26234207). However, it is possible that GLTP deletion might indirectly affect vesicular trafficking, for example, by disrupting ER morphology and function.

Nevertheless, we agree that it important to explore possible function in vesicular trafficking in oligodendrocytes. To address this question, we performed proteomics analysis of myelin purified from P28 GltP cKO and Cre control mice. The quality of the samples was validated by principal component analysis, cellular component GO term enrichment analysis and GltP reads. Enrichment analysis indicated upregulation of pathways such as neurodegeneration and lipid droplet organization in GltP cKO, consistent with our electron microscopy and lipidomics findings. Two lipid transfer proteins were upregulated in cKO: Mospd2 and C2cd2l. The data is shown in a new Supplementary Figure 5c-h (see also below).

Our proteome and lipid analyses enabled us to test for disruptions in vesicular trafficking, which would manifest as a depletion of membrane proteins from myelin. However, most transmembrane proteins did not show significant changes, and overall, there was no pattern of decrease, contradicting the hypothesis, that GLTP directly affects vesicular trafficking (see Supplementary Fig5e-g). Instead, our lipid analysis revealed that HexCer constitutes the majority of downregulated lipids. Thus, we conclude that GLTP influences lipid trafficking to

myelin. Because non-canonical functions of this protein cannot be excluded, we have now mention this in the discussion.

(Figure and legend above for reviewer#1 point#5)

In addition to this major weaknesses, below are some additional suggestions that may help to improve this study.

1. Tubular ER is present in most of the animal cells. The existence of tubular ER in myelin sheath suggests tubular ER can support myelin growth, but it does not explain why oligodendrocytes, not other brain cell types, can produce enormous amount of lipids. One possibility is ER makes more contact with plasma membrane in oligodendrocytes. A quantification of tubular ER-myelin sheath coverage will help to clarify this issue. Also, a comparison of several tubular ER lipid transport gene expressions among different brain cell types will tell us if oligodendrocytes have more ER lipid transport machinery. Please also clarify whether tubular ER in inner tongue is still connected with ER network in the cell body.

This is very good suggestion, we have now compared the ER-myelin contact sites to ER-axon contact sites. We found that the ER has much closer contac to plasma membrane t in the inner tongue of myelin compared to ER found in axon. The data is presented in a new Fig 1h (see below).

We have now also added an analysis showing the expression level of the known lipid transfer proteins (based on the RNA-Seq dataset of Zhang et al.). GLTP stands out as the most highly expressed lipid transfer protein in oligodendrocytes. NPC1 and 2 are also expressed at relatively high levels, but these proteins operate in lysosomes. The analysis has been added to Supplementary Figure 3a (see also below). However, lipid transfer proteins are generally expressed at low levels, but despite their low expression, they can have important functions. Future research using proximity labeling for proteomics is necessary to isolate the lipid transfer machinery in oligodendrocytes. The reviewer's final point suggests reconstructing the tubular ER from the inner tongue through the myelin and related processes to the cell body. We acknowledge that this would be an intriguing analysis. However, despite our expertise in 3D-EM, this task is too challenging and impossible to perform reliably within the time constraints of a revision.

Gene Expression Profiles Involved in Nonvesicular Lipid Transport (RNA-seq from Zhang et al. 2014)

2. Page 6, line 1: The authors observed a decrease in cell size after knocking down Rtn4 and concluded that tubular ER is crucial in myelination. It is more appropriate to use “membrane addition” instead of “myelination” as there is no myelination in their in vitro model.

We now used membrane expansion, as suggested. In addition, we now quantified MBP positive sheet size as a more accurate read-out. We found a reduction in MBP positive sheet size. The data has been added to Supplementary Figure 2g (see also below).

3. Page 8 line 1 and figure 5: They found hypomyelination and axon degeneration in Gltp knockout mice. As axon degeneration and demyelination often happen together, whether the observed “hypomyelination” phenotype is due to developmental hypomyelination or due to demyelination is not clear. Please clarify this. Also, please quantify the ratio of dysmyelination (abnormal myelination pattern) if there is any.

The reviewer addresses an important point whether the myelin phenotype could be explained non-autonomously by axon degeneration. To address this point we estimated the number of axons, but found no differences. Our finding of abnormally thinly myelinated axons as shown by the g-ratios analysis of WT and GLTP KO at P28, is therefore consistent with reduced formation of myelination. In addition, as requested by the reviewer we quantified the number of myelin alteration of which outfoldings. There were no differences between WT and GLTP KO. However, we find increased number of myelin whorls, which is another common feature of dysmyelination. Because our analysis was performed in development, we now use the term dysmyelination and not degeneration.

The data has been added to Supplementary Figure 5a, b (see also below).

4. Page 14, figure 1g: Are the ERs in the inner tongue closer to the inner tongue plasma membrane facing the axon or the inner tongue plasma membrane facing outer layers of myelin? Do the author envision lipid transfer to occur on the side facing the axon or the side facing outer myelin layers? Please also include how they define the distance between myelin and tubular ER in the method section. Is it the closest distance between tubular ER and tubular?

Our analysis did not differentiate between the ER contacting the axon-facing inner tongue or the outer myelin layer. Since membranes are fluid and lateral diffusion rates are relatively fast, we believe that adding lipids to either side will result in the expansion of the leading edge at the inner tongue. As requested, we have now updated the methods section to specify how the analysis was conducted.

5. Page 16, figure 2: they found tubular ER markers, REEP5 and RTN4 are dramatically reduced in 6 months old mice as compared to P14. Are there fewer tubular ER in the myelin sheath of 6 months old mice as well.

We have now conducted an analysis of tubular ER in adult myelin sheaths. There is an almost complete absence of inner tongue and associated ER in adult myelin. The data has been added to Supplementary Figure 2d, e (see also below).

6. Page 17, figure 3: the intensities of GLTP seems stronger in the cell body than in branches. Please provide the quantification of cell body/process ratio of GLTP as they did in figure 2h. This information could be helpful for understanding whether GLTP mostly functions in the cell body or processes. Also, does Gltp knockout affect ER morphology in the cell body?

We have now conducted a quantification of GLTP in the cell body versus the processes, and compared this to KDEL, a protein primarily localized to ER sheets. We found a higher

processes to cell body ratio for GLTP compared to KDEL. The data has been added to Fig 3g, h (see also below).

The reviewer also asks whether GltP affect ER morphology in the cell body. We indeed found evidence that this is the case. The data is found in Supplementary Fig 4e,f

7. Page 18, figure 4: In figure 4a and 4c, although ER morphology is changed in GltP knockout mice, it is unclear whether the distance between ring ER and myelin plasma membrane is affected. A quantification would be informative.

We have now quantified the distance of ER to myelin. We found that distance is very similar to WT mice. However, please note that rings are big and often fills the entire inner tongues. Thus, they cannot escape being at close distance to the membrane.

The data has been added to Supplementary Figure 4g (see also below).

8. Page 20, figure 6g: from the image they provided, it looks like there are more star/arborized shaped cells in Gltp KO group. This may suggest that Gltp KO also affects the maturation of oligodendrocytes. Please quantify the number of oligodendrocytes at different maturation stages based on their morphology.

We have now quantified the morphology of oligodendrocytes and classified them into star/arborize cell shape versus sheet shaped. We find that GLTP KO leads to a higher number of star/arborize cell shaped oligodendrocytes.

The data has been added to Supplementary Figure 6q-s (see also below).

9. Besides citing several references, it is unclear how ER was identified and confirmed on EM. How can the authors be certain that those membranous structures are ER and not Golgi, endosomes, lysosomes, or other compartments?

We have now updated the methods section to provide a clear description which criteria were used to define the ER by EM:

ER membranes in ultrastructural data were identified according to morphological criteria. The tubular network structure in the volume data sets distinguishes ER membranes from endocytotic or other vesicles (Heinrich et al. Nature volume 599, pages141–146 (2021)). While different staining protocols result in diverging grey values of organellar membranes, in several rOTO protocols ER membranes appear darker in comparison to the plasma and other

organellar membranes by TEM and SEM backscattered detection (Terasaki et al. J Cell Sci (2018) 131 (4): jcs210450.). Additionally, in our datasets, the Golgi and endocytic compartments have lighter lumens compared to the ER. Another feature of ER membranes that we observe in oligodendrocyte processes is their narrow opposition of membranes with little lumen, comparable to axonal ER (Terasaki et al. J Cell Sci (2018) 131 (4): jcs210450.).

10. Another suggestion is to perform live imaging to show the dynamics of ER and plasma membrane contact events in oligodendrocytes. FRET assays can be considered

This would certainly be an intriguing analysis. Unfortunately, this experiment is not feasible. We have conducted live imaging in zebrafish and attempted it in primary oligodendrocytes, but the resolution achieved with light microscopy is far too low to draw any conclusions about the dynamics of contact sites.

Reviewer #3 (Remarks to the Author):

In this study, the authors use volume electron microscopy to visualize the ER during myelination. The authors demonstrate that the tubular ER is present in the inner layer of myelin where it is growing. They also explore the role of glycolipid transfer protein (GLTP) in myelination with the use of a conditional Gltp knockout mouse strain. They show that loss of Gltp leads to defects in the ER, myelin formation, and levels of HexCer – a major myelin lipid. These findings are also validated in primary OPC cultures.

The study is very well written and the conclusions are well justified by the data shown. The findings reveal important new insights into how lipids are installed into newly forming myelin membranes and the roles of the ER and GLTP in this process.

We thank the reviewer for the supportive comments and helpful suggestions made.

Some minor concerns and questions are noted below.

p. 3 – How are the ER organelles definitively distinguished from other types of organelles?

We have now updated the methods section to provide a clear description which criteria were used to define the ER by EM:

ER membranes in ultrastructural data were identified according to morphological criteria. The tubular network structure in the volume data sets distinguishes ER membranes from endocytotic or other vesicles (Heinrich et al. Nature volume 599, pages141–146 (2021)). While different staining protocols result in diverging grey values of organellar membranes, in several rOTO protocols ER membranes appear darker in comparison to the plasma and other organellar membranes by TEM and SEM backscattered detection (Terasaki et al. J Cell Sci (2018) 131 (4): jcs210450.). Additionally, in our datasets, the Golgi and endocytic compartments have lighter lumens compared to the ER. Another feature of ER membranes that we observe in oligodendrocyte processes is their narrow opposition of membranes with

little lumen, comparable to axonal ER (Terasaki et al. J Cell Sci (2018) 131 (4): jcs210450.).
The ER-plasma membrane.

p. 4 – In results section, should specify that the “white matter” in Fig. 2d/f is from spinal cord

We specified.

p. 6 – could specify what cells CNP-Cre targets

We specified.

p. 8 – From reading the methods sections, the lipidomics methodology is a targeted approach – not an untargeted approach. Not distinguishing between GalCer and GlcCer does not make it untargeted.

We corrected

p. 8 - are lipids really “downregulated and upregulated” in the same way as genes and proteins? – I would argue that this should just be “increased and decreased”

We agree, and corrected accordingly.

p. 8 – how is the MS method distinguishing the individual FA species and why are they not identified for the HexCer species? I found the answer in the referenced methods paper, but it should perhaps be included here as well.

It is important to note, that the presented FAs are not free FAs, but are part of the complex lipids, such as PLs. The FA species are identified by fragmentation of the precursors, producing FA fragments. This is done for PLs, DAGs, TAGs, CE, but not for HexCer, SM, and other SLs as they fragment only poorly and don't easily provide the FA (or LCB) information in our method. We now added this information.

p. 8 – Is the PCA (Fig. 6f) performed on the individual lipids in Fig. 6a-c or on the total levels shown in Fig. 6e?

PCA at Fig 6f is for Fig 6a-c; PCA at supplementary Fig 6b is for Fig 6e

p. 8 – All of the lipid data is reported as mol%, which is a normal/standard way of reporting these values. Is this normalized by total moles of lipid measured by mass spec? Or by total lipid extract weight? Are there differences in the total lipid levels between the cKO and cre-only? Have you normalized by protein amount or tissue weight? This could provide a different view of the dataset.

The reported value is % of total moles, will make it clear in the text. Total lipid counts (moles) are not significantly different.

The mol% values are normalized to total moles of lipid measured by mass spec. The amount of input material (as wet weight) is provided in the dataset. The reviewer is right, it could provide a different picture, esp. when there is for example a strong increase in TAGs under certain conditions (say from 10 to >50 mol%). This might then result in lower mol% values for all the other classes. However, if the lipid class levels are rather stable across conditions, then mol% presentation of the data is a good choice.

Fig. 6 – the caption mentions that the raw lipidomics data is in the supp files, but I did not see it.

Will upload the supplementary table with the data.

Response to the reviewers' comments

We thank all three reviewers for the constructive feedback in improving the manuscript.

Reviewer #1 (Remarks to the Author):

I am pleased to see the revised manuscript and the adjustments that have been made for the story to be more cohesive. I deeply appreciate the meticulous actions taken by the authors in addressing all reviewer suggestions and points. The authors justified the use of REEP5 and RTN4 as these are the most abundantly expressed tubular ER markers in oligodendrocytes. The authors addressed my points further, especially with the inclusion of new tubular ER marker expression, IHC/volume EM of other reticulons, and quantification of the MBP sheet in primary oligodendrocyte cultures after tubular ER protein knockdown, which supports their claim that tubular ER is present and effects myelination. The author's rationale was strengthened with the addition of gene expression profiles in nonvesicular lipid transport, which demonstrates GLTP as the most abundant lipid transfer protein in oligodendrocytes, supporting GLTP as a strong target mechanism for the transfer of non-vesicular lipids. It's also greatly appreciated for the distinction of the term dysmyelination instead of degeneration, as well as the clear description given for the identification of ER in EM. The additional layer of transcriptomic profiling analysis and quantification of dysmyelination profiles under GLTP deletion greatly contributed to understanding the large impact GLTP deficiency has on myelination and showed the upregulation of neurodegeneration and other supplemental pathways. The authors have also included a section in the discussion to address potential compensatory mechanisms in response to the fluctuations of other myelin lipid components under GLTP deletion. Lastly, mentioning the non-canonical functions of GLTP improve the discussion and give a well-rounded view on the topic for the audience. The revision has improved immensely with all actions taken from the authors, and I believe these changes significantly improved the manuscript and is an overall interesting read.

Reviewer #2 (Remarks to the Author):

The authors have addressed all my concerns. I recommend the publication of this manuscript.

Reviewer #3 (Remarks to the Author):

The authors have appropriately addressed all of the concerns raised in the initial review. I would recommend that this important study be published.